# An Approach for Selecting the Most Explanatory Features for Facial Expression Recognition

**Pedro D. Marrero-Fernandez** [1], **Jose M. Buades-Rubio** [2,*], **Antoni Jaume-i-Capó** [2,3] **and Tsang Ing Ren** [1]

1  Centro de Informática, Universidade Federal de Pernambuco (UFPE), Recife 50670-901, Brazil; pdmf@cin.ufpe.br (P.D.M.-F.); tir@cin.ufpe.br (T.I.R.)
2  Computer Graphics, Vision and Artificial Intelligence Group (UGIVIA), Department of Mathematics and Computer Science, University of the Balearic Islands, 07122 Palma, Spain; antoni.jaume@uib.es
3  Laboratory of Artificial Intelligence Applications (LAIA@UIB), University of the Balearic Islands, 07122 Palma, Spain
*  Correspondence: josemaria.buades@uib.es; Tel.: +34-971-17-2943

**Abstract:** The objective of this work is to analyze which features are most important in the recognition of facial expressions. To achieve this, we built a facial expression recognition system that learns from a controlled capture data set. The system uses different representations and combines them from a learned model. We studied the most important features by applying different feature extraction methods for facial expression representation, transforming each obtained representation into a sparse representation (SR) domain, and trained combination models to classify signals, using the extended Cohn–Kanade (CK+), BU-3DFE, and JAFFE data sets for validation. We compared 14 combination methods for 247 possible combinations of eight different feature spaces and obtained the most explanatory features for each facial expression. The results indicate that the LPQ (83%), HOG (82%), and RAW (82%) features are those features most able to improve the classification of expressions and that some features apply specifically to one expression (e.g., RAW for neutral, LPQ for angry and happy, LBP for disgust, and HOG for surprise).

**Keywords:** facial expression recognition; explain learning; explainable artificial intelligence; ensemble of classifiers; sparse representation; multiple classifier systems

## 1. Introduction

Computers are quickly becoming a ubiquitous part of our lives, and we spend a large amount of time interacting with computers of one type or another; however, the devices that we use are indifferent to our affective states, as they are emotionally blind. Successful human–human communication relies on the ability to read affective and emotional signals. Human–computer interaction, which does not consider the affective states of its users, loses a large part of the information available from the interaction.

At present, expression recognition can be achieved using neural networks. A neural network behaves like a black-box model; therefore, it is difficult to understand how it performs its classifications. Recent works have attempted to create attention maps to understand the results obtained by neural networks [1,2]; however, these maps do not produce reasonable interpretations of the types of features learned by the neural network.

Other solutions, such as sparse representation, help in studying the classification performance. In this context, a trustworthy intelligent system must be able to explain its decisions and actions to human users through the use of techniques that produce more understandable models, while maintaining high performance levels [3].

Thus, in this paper, we analyze the importance of each feature used for facial expression recognition. Feature-relevance techniques seem to be some of the most-used schemes

for post hoc explanation [4,5], as they can provide an explicit description of the inner behavior of the model, contributing to the goal of designing an explainable intelligent system.

In our work, we focus on determining which feature sub-spaces are the most advantageous, where sparse representation (SR) was used to determine the features.

The remainder of this paper is organized as follows: Section 2 introduces the related work. Section 3 describes the methodology, local and geometric features, expression classification, and combination framework. Section 4 describes the specific experimental process. Section 5 analyzes and discusses the results. The research is summarized, and our conclusions are offered in Section 6.

## 2. Related Work

Guidotti et al. [6] carried out an exhaustive literature review to explain learning systems. In [7], Letham et al. generated an explanation using rules and Bayesian analysis by making if/else decisions. LIME was proposed as a solution in [8], with the aim of explaining the predictions of any model in an interpretable manner.

Chandrashekar et al. [9] provided an introduction to feature-selection techniques. They concluded that "more information is not always good in machine learning applications". This is an important issue, as it means that adding more features does not always elicit better results. In [10], Weitz et al. used LIME and layer-wise relevance propagation (LRP) to explain how a neural network can distinguish between pain, happiness, and disgust.

In [11], Gund et al. used videos from CK+ and a temporal convolutional network to determine the most important facial landmarks (i.e., 68 OpenFace landmarks) that can be used to infer facial expressions. In [12], Ter Burg concluded that human participants found the explanations for the geometric feature-based DNN better overall than the Grad-CAM explanations for the CNN. In Wang et al., $M^2$Lens [13] contributes to visualizing and explaining multi-modal models for sentiment analysis, which is related to facial expression recognition.

Lian et al. [14] and Kim et al. [15] determined the regions used for the classification of individual expressions. Deramgozin et al. [16] determined how a CNN classifies different expressions, both locally and globally. However, none of the reviewed works have determined which characteristics are the most important for better facial expression recognition. Zhu et al., in their recent publications, have attempted to focus the majority of their processing on a target region of interest, using a so-called visual attention mechanism [1,2]. Although the existing attention methods have contributed greatly to facial expression recognition, according to Bonnard et al. [2], there is a problem associated with the insufficient utilization of spatial features. In our opinion, there is a need to begin to understand the different types of features that contribute to face-recognition analysis, which could ultimately improve the actual systems in practice.

Several papers related to feature extraction for facial expression recognition (FER) can be found in the literature. Ying et al. [17] used local binary patterns (LBP) and Image Raw (RAW) to train two classifiers using a sparse representation-based classifier (SRC). For each of these schemes (LBP + SRC and RAW + SRC), the approximation error signal was obtained for each class. This error was used as a fuzzy measure for the evaluation of a decision rule. The residual ratio was calculated as the ratio between the second smallest residual and the smallest residual for the LBP + SRC and RAW + SRC methods. If the results for the two classifiers were not the same, the classification with the larger residual ratio was chosen. In the study by Li et al. [18], the classifiers were trained using local phase quantization (LPQ) and Gabor wavelets + Adaboost (GW). The Adaboost algorithm was used to select the most effective 100 features from each Gabor filter. As reported in [17], the classification result with the larger residual ratio was chosen if the classification using the two classifiers differed [17]. Ouyang [19] used a histogram of oriented gradients (HOG) and LBP. This approach was based on the idea that features are complementary, as HOG mainly extracts a contour-based shape, while LBP primarily extracts the texture information of the images. Kittler et al. [20] showed that the output of each classifier could be used to evaluate

a decision rule, then they applied combination rules. Of these, the product rule (PR) and the SR provided the best results. Ptucha et al., Ji et al., and Tsalakanidou et al. have also employed dynamic features in [21–23]. In these works, the variability of facial changes was studied using regions or points of interest in the face images.

The notion of SRs—finding sparse solutions to under-determined systems—has been applied in several scientific fields. Olshausen et al. used sparse models that are similar in nature to the network of neurons in V1—the first layer of the visual cortex in the human brain—and, more generally, to the mammalian brain [24,25]. Patterns of light are represented by a series of innate or learned basis functions, whereby sparse linear combinations form surrogate input stimuli to the brain. Similarly, for many input signals of interest, such as natural images, a small number of exemplars can form a surrogate representation of a new test image.

In SR systems, new test images are efficiently represented by sparse linear coefficients on a dictionary $D$ of over-complete basis functions. Specifically, SR systems comprise an input sample $x \in R^m$ (where $m$ is the number of features), along with a dictionary $D$ of $n$ samples, $D \in R^{m \times n}$. SR solves the coefficients $\alpha \in R^n$ that satisfy the $l_1$ minimization problem $x^\star = D$.

Wright et al., Weifeng et al., and Zhen et al. showed the advantages of exploiting sparsity in pattern classification, which has been demonstrated extensively for FER problems [26–28]. The experimental results of [26] showed that the magnitude of the representation error in the facial feature vectors obtained by SR provided a good metric with which to classify facial expressions.

Different expression representation techniques have been created, and many more are expected to be created in the next few years. Representations may be features designed by experts (feature engineering methods), as was demonstrated by Ptucha et al., Mollahosseini et al., and Zhang et al. [21,29–31], or embedded vectors obtained from training a deep neural network. The main objective of this work was to define a new facial expression recognition system that uses the representations obtained from different sources and combines them using a learned model. For this goal, we performed the following steps: (1) we first obtained several representations of facial expressions by applying different representation methods; (2) we then transformed each obtained representation into the same domain, the domain of representation errors of sparse representation (SR); finally, (3) in this new space, we trained a combination model to classify the signals. We used the extended Cohn–Kanade data set (CK+), the BU-3DFE data set, and the JAFFE data set for validation. The major innovation of this work is the exploration of feature extractors to select those features that are the most explanatory for facial expression recognition, using a sparse representation-based classifier.

## 3. Methodology

In this section, we detail the feature extraction and expression classification methods. The feature extraction methods used can be grouped according to local and geometric features, as well as global features extracted from pre-trained deep models. Local features use neighbor information, while geometric features consider points of interest in the face. Pre-trained neural network models provide a $k$-dimensional feature vector extracted from the last layer. Figure 1 represents the scheme of the proposed approach.

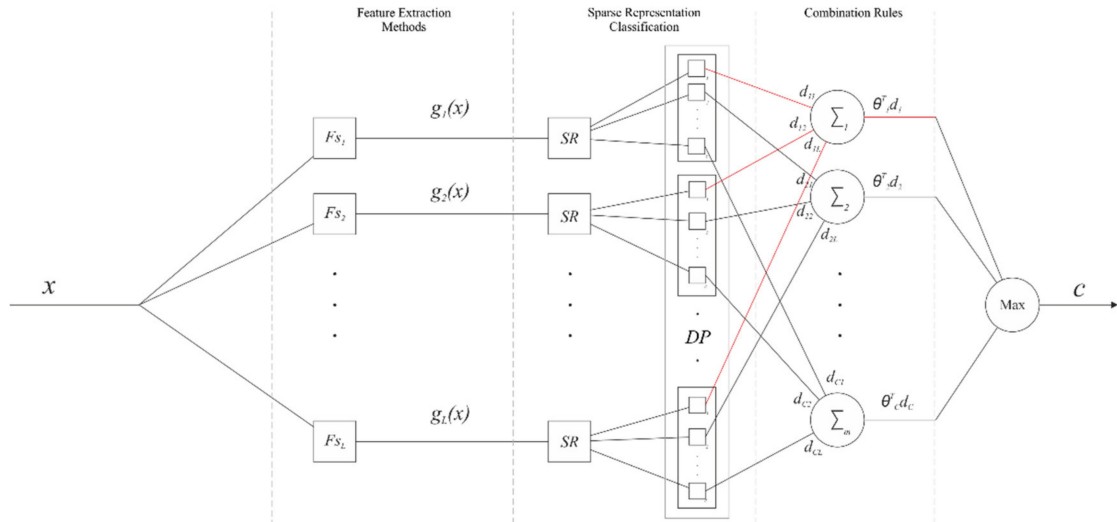

**Figure 1.** Proposed classifier system, divided into three sections: feature extraction methods, sparse representation classification, and combination rules.

### 3.1. Feature Extraction Methods

In this work, we group the feature methods into local, geometric, and global. The local features were defined through the extraction of supervised features from facial patches. The facial patches were defined as regions in the face that were active during different facial expressions. It has been reported that some facial patches are common during the production of all basic expressions, while others are confined to a single expression [32]. The results indicated that these active patches were positioned below the eyes, between the eyebrows, and around the nose and mouth. To extract these patches from the face image, we first located the facial components. The locations of active patches were defined with respect to the positions of landmarks, which were estimated using Open-Face [33]. Happy and Routray [34] have observed that features formed from fewer facial patches can replace high-dimensional features without a significant decrease in recognition accuracy. Geometric features were defined according to the distances and regions between the different landmarks [23]. The global methods employed unsupervised features obtained by the pre-trained deep learning models, VGG and VGGFace, over the entire image [35].

#### 3.1.1. Local Features

The local features were established through the extraction of supervised features from facial patches, defined as the regions of the face that are active during different facial expressions. We used the following methods for the extraction of local features:

- Gabor wavelet filters (GW): This technique has previously been applied successfully for facial expression recognition [18,30]. In our work, 68 landmark points were selected. For each point, a patch sized $(2k+1) \times (2k+1)$ was used to compute the feature vector. Four scales and eight orientations were used to calculate the Gabor kernels, where $k = 7$. This selection generated a vector of 2176 ($68 \times 4 \times 8$) elements.
- Local binary patterns (LBP): this system has been widely used as a robust illumination-invariant feature descriptor [19,27,36,37]. This operator generates a binary number by comparing neighboring pixel values with the center pixel value. The uniform LBP and rotation-invariant uniform LBP [38] were also used in the experiment. These methods generated a vector that was the same size as the image ($256 \times 256$).
- Histograms of oriented gradients (HOG): This concept has also been successfully applied in facial expression recognition [19,39]. The basic idea of HOG features is that the local object's appearance and shape can often be well-characterized by the distribution of local intensity gradients or edge directions, even without precise knowledge of the corresponding gradient or edge positions. The orientation analysis is

- robust to changes in illumination, as the histogram possesses translational invariance. The associated vector size was $256 \times 256 \times 2$.
- Local phase quantization (LPQ): The LPQ feature [18,40] is a blur-robust image descriptor. The LPQ descriptor is based on the intensity of low-frequency phase components with respect to a centrally symmetric blur. Therefore, LPQ employs the phase information obtained from the short-term Fourier transform, which is locally computed on a window around each pixel of the image. The vector size was $256 \times 256$.
- RAW: The image intensity has also been used as a feature vector (RAW) [17,41]. Here, the vector size was $256 \times 256$.

### 3.1.2. Geometric Features

Geometric measurements (GEO) were computed using 68 landmarks [23]. Figure 2 shows the order of the landmarks selected in our work. The order and number of landmarks varied in some of the previous works in the literature. We maintained the appropriate correspondence in all cases. Table 1 details the 17 geometric distance measures used.

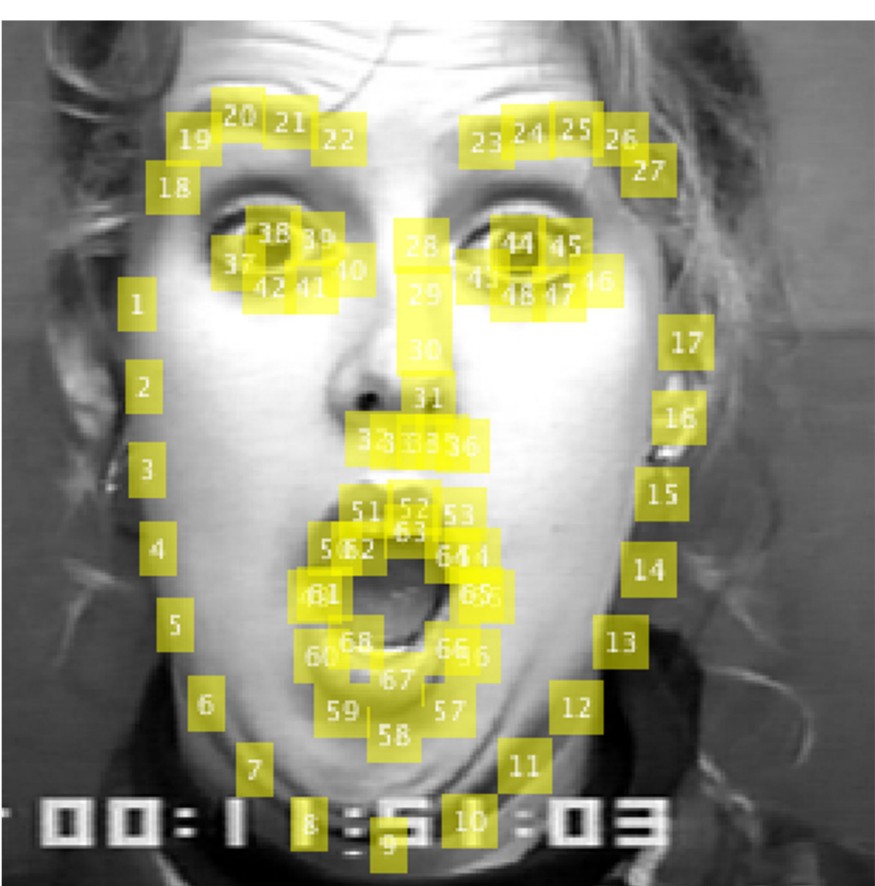

**Figure 2.** The 68 landmarks used in our work. Each landmark is marked with its number. Original image extracted from BU-3DFE [42,43].

We also used shape measurements for the eye, nose, and mouth regions. We defined these regions using the landmarks in each region. For example, the left eye region is defined by landmarks 37, 38, 39, 40, 41, and 42. Measures were defined for each region, as follows: solidity ($M_{18}$) returns a scalar specifying the proportion of the pixels in the convex hull that are also in the region, computed as $Area/ConvexArea$; the axes relationship ($M_{19}$) is the ratio between the lengths of the minor and major axes of an ellipse, which has the same normalized second central moments as the region, computed as $AxisMin/AxisMax$; tshe circularity factor ($M_{20}$) is computed as $4\pi Areas/Perms^2$; eccentricity ($M_{21}$) is a scalar that specifies the eccentricity of the ellipse, which has the same second moments as the region;

and extent ($M_{22}$) is a scalar that specifies the ratio of pixels in the region to pixels in the total bounding box. The distances of the centroids of each region to the center of the nose were also calculated. Table 2 summarizes the vector size used for each feature; the total vector size was 329,992.

**Table 1.** Geometric facial measurements. $d_{ij}$: the Euclidean distance between landmarks $i$ and $j$; $\alpha$: the angle between two lines; $\xi_{ij}$: the line defined by $i$ and $j$; $l_{ijk}$ the length of the curve defined by $i$, $j$, and $k$; $o_1$ and $o_2$: the centers of the right and left eye, respectively.

| Measurement | Name | Distances Involved |
|:-----------:|:----:|:------------------:|
| $M_1$ | Inner eyebrow displacement | $d_{40,22}, d_{43,23}$ |
| $M_2$ | Outer eyebrow displacement | $d_{18,o_1}, d_{28,o_2}$ |
| $M_3$ | Inner eyebrow corners dist. | $d_{22,23}$ |
| $M_4$ | Eyebrow from nose root dist. | $d_{22,28}, d_{23,28}$ |
| $M_5$ | Eye opening | $d_{39,41}, d_{44,48}$ |
| $M_6$ | Eye shape | $d_{39,41}/d_{37,40}, d_{44,48}/d_{29,46}$ |
| $M_7$ | Nose length | $d_{29,31}$ |
| $M_8$ | Nose width | $d_{32,36}$ |
| $M_9$ | Lower lip boundary length | $l_{55,56,57,58,59,60,49}$ |
| $M_{10}$ | Mouth corners dist. | $d_{49,55}$ |
| $M_{11}$ | Mouth opening | $d_{63,67}$ |
| $M_{12}$ | Mouth shape | $d_{63,67}/d_{49,55}$ |
| $M_{13}$ | Nose–mouth corners angle | $\alpha(\xi_{32,49}, \xi_{36,55})$ |
| $M_{14}$ | Mouth corners to eye dist. | $d_{o_1,49}, d_{o_2,55}$ |
| $M_{15}$ | Mouth corners to nose dist. | $d_{49,34}, d_{55,34}$ |
| $M_{16}$ | Upper lip to nose dist. | $d_{52,34}$ |
| $M_{17}$ | Lower lip to nose dist. | $d_{67,34}$ |

**Table 2.** Feature summary. For each feature, we provide its type and size.

| Type | Feature | Vector Size |
|:----:|:-------:|:-----------:|
| Local | GW | 2176 |
| Local | LBP | $256 \times 256$ |
| Local | HOG | $256 \times 256 \times 2$ |
| Local | LPQ | $256 \times 256$ |
| Local | RAW | $256 \times 256$ |
| Geometric | GEO | $68 \times 2$ |

### 3.1.3. Pre-Trained Deep Models

Several deep learning algorithms have been proposed and applied to FER [29,44,45], particularly from an explainable artificial intelligence (XAI) point of view [46–48]. Our interest was in the combination of different representation spaces. Thus, we selected pre-trained models to obtain facial expression representations. The training of deep learning architectures for FER poses a problem when using data sets such as CK+, JAFFE, and BU-3DFE, due to the small number of elements in these sets. Models trained with such data sets are prone to overfitting, which can hinder objective analysis when determining the contribution of the characteristics of the model to the system. Therefore, we selected general classification models (i.e., object classification and facial classification models) that were pre-trained on extensive databases, including the VGG-face [35] and VGG [49] models. In the case of the VGG-face model, as a feature, we selected the output of the Rectifier Linear Unit (ReLU) layer 33. For the VGG model, we selected the output of the ReLU layer 34. The size of both vectors was 4096.

### 3.2. Expression Classification

Consider a set of training signals $D = [D_1, D_2, \ldots, D_k] \in R^{m \times p}$ from $k$ different classes, where the columns of each sub-matrix $D_j = \left[d_1^j, d_2^j, \ldots, d_{n_j}^j\right] \in R^{m \times n_j}$ are signals from the class $w_j$. Ideally, we have sufficient training samples of class $w_j$, such that a test signal

$x \in R^m$ that belongs to the same class can be approximated by a linear combination of the training samples from $D_j$, which can be written as:

$$x = \sum_{i=1}^{n_j} \alpha_i^j d_i^j, \tag{1}$$

which can be rewritten as $x = D\delta_j(\alpha) \in R^m$, where $\delta_j(\alpha) = \left[0, \ldots 0, \alpha_1^j, \alpha_2^j, \ldots, \alpha_{n_j}^j, 0 \ldots, 0\right]^T \in R^p$ is a vector of coefficients having many values equal to zero, except for those associated with the class $w_j$. As a valid test sample, $x$ can be sufficiently represented only using the training samples from the same class; this representation is the sparsest among all others, finding the identity of $x$ is equal to finding the sparsest solution of Equation (2). This is equivalent to solving the following optimization problem ($l_0$-minimization):

$$\alpha^\star = arg \min_{\alpha \in R^p} \|\alpha\|_0 \text{ s.t. } D\alpha = x. \tag{2}$$

However, solving the $l_0$ minimization of an undetermined system of linear equations is NP-hard. If the sought solution $\alpha^\star$ is sparse, the solution of the $l_0$ minimization problem, as defined in Equation (2), is equal to the solution of the following $l_1$ minimization problem [50]:

$$\alpha^\star = arg \min_{\alpha \in R^p} \|\alpha\|_1 \text{ s.t. } D\alpha = x. \tag{3}$$

Then, the estimate $x$ using the coefficients corresponding to a given class $w_j$, $x \approx \hat{x}_j = D\delta_j(\alpha^\star)$, is possible. This is consistent with the previous findings, and also mimics the behavior of simple cells in the visual cortex. The error of the representation, $e_j = |x - \hat{x}_j|$, can be used to determine the class of the signal $x$.

### 3.3. Proposed Combination Framework

When different classifiers based on SR are considered, the reconstruction error can be used to evaluate the combination rules. This supposes that the probability of success of each classifier $D_i$ for each class, $P(w_j|D_i)$, is the same. Combination methods of this type are called "class-conscious" [51].

Depending on its intrinsic characteristics, each expression can best be characterized in a particular subspace or subset. For example, expressions that involve some form of facial movement (e.g., opening the mouth) may be better described by the shape spaces, which are recorded as changes in gradient. On the other hand, changes in the frequency intensity of the image may be best characterized by texture analysis methods. This property suggests that classifiers exist that are specialists (experts) for some classes. As such, these classifiers should have greater weight in the final decision regarding classification.

For the calculation of the weights, a new feature space based on the output of each of the classifiers was generated. The variable $d_{i,j}(x)$ denotes the support that classifier $D_i$ gives to the hypothesis that $x$ comes from class $w_j$. The larger the support, the more likely it is that the class label belongs to $w_j$. In this approach, $d_{i,j}(x)$ are features in a new space, defined as the intermediate feature space [51]. The support for class $w_j$ is calculated as:

$$\mu_j(x) = \sum_{i=1}^{L} \theta_{i,j} d_{i,j}(x). \tag{4}$$

Linear regression is the most commonly used procedure to derive the weights for this model [21]. Algorithm 1 describes each step in the classification of a test pattern. Note that the output of each classifier $d_{i,j}$, for the class $w_j$, creates a new feature space. For each output subspace, an estimated model $\theta_j$ weighs the decision of each classifier in the class $w_j$. For the experiments, the fuzzy integral (FI) [51], linear opinion pools (LOP) [52], SVM, and naïve Bayes method were used to adjust the output of the classifiers (see Equation (8)). Other techniques for combining experts exist, but they need to be trained using a large number of samples [53].

---

**Algorithm 1.** Sparse representation fusion classification (SRFC). $D_1, \ldots, D_L$, with $D_i \in R^{n_i \times m}$, are the dictionaries for each feature space; $g_1(x), \ldots, g_L(x)$, with $g_i(x) \in R^{n_i}$ and $g_i(x) \neq g_j(x)$, are the feature extraction methods

---

**1: Calculate sparse representation:**
For $i = 1, \ldots, L$

$$\alpha_i^\star = \arg \min_{\alpha \in R^p} \|\alpha\|_0 \, s.t. \, g_i(x) = D_i \alpha \tag{5}$$

**2: Calculate the vote of each representation to each class:**
For $i = 1, \ldots, L$ and $j = 1, \ldots, C$

$$r_{i,j} = \|g_i(x) - D_i \delta_j(\alpha_i^\star)\|_2^2, \tag{6}$$

where $\delta_j$ selects the entries of $\alpha^\star$ corresponding to the class $j$, and $r_j$ represents the residual test sample $g_i(x)$ with the linear combination $D_i \delta_j(\alpha^\star)$. To obtain the vote for each class, the *softmax* function is applied to the inverse of the normalization of $r_{i,j}$:

$$d_{i,j} = \sigma_{softmax}\left(1 - \frac{r_{i,j}}{\|r_{.,j}\|_1}\right), \tag{7}$$

where $r_{.,j}$ refers to the column $j$ as a vector and $d_{i,j}$ represents the decision profile.

**3: Trained combination rules:**

$$\mu_j(x) = \sum_{i=1}^{L} \theta_{i,j} d_{i,j}(x). \tag{8}$$

The weights $\theta_j$ are estimated for the decision profile for class $w_j$.

**4: Classification:**

$$\hat{j} = \arg \max_{j \in 1, \ldots, C} \mu_j. \tag{9}$$

**5: Return**
The estimated class $\hat{j}$ for the signal $x$

---

## 4. Experiments

In order to determine the influence of different features, we performed four experiments. (1) Statistical analysis was employed to determine which combination of rules presented the best classification results, for which purpose we evaluated 247 combinations of feature extraction methods—i.e., we considered all possible combinations of 8 features, excluding empty sets and those using only one feature set ($2^8 - 1 - 8 = 247$)—and 14 combination rules in three different data sets. (2) To investigate the generalization performance of the proposed method vs. individual classification methods, we performed an inter-database experiment. (3) We analyzed the influence of the methods for each class; and (4) we compared the obtained results with those using state-of-the-art methods on the same data set and experimental protocol. The results obtained from these experiments are described in detail in the next section.

### 4.1. Data Set

To evaluate our proposed method, we used three public data sets: the extended Cohn–Kanade (CK+) [54], JAFFE [55], and BU-3DFE [43]. In all experiments, person-independent FER scenarios were used [56]. Therefore, the subjects in the training set were completely different from the subjects within the test set (i.e., the subjects used for training were not used for testing). Following the recommendations in [38], each face image used in our experiments was cropped, based on the locations of the eye. The landmarks were obtained using Open-Face. The cropped face images were rescaled to 256 × 256 pixels. Figure 3 shows the various classification results.

- CK+ data set: This includes 593 image sequences from 123 subjects. From the 593 sequences, we selected 325 sequences of 118 subjects, each of which met the criteria for one of the seven expressions [54]. The selected 325 sequences consisted of 45 angry, 18 contempt, 58 disgust, 25 fear, 69 happy, 28 sadness, and 82 surprise sequences [54]. In the neutral face scenario, we selected the first frame of the sequence of 33 randomly selected subjects.
- BU-3DFE data set: This is known as the most challenging and difficult data set, mainly due to the presence of a variety of ethnic/racial ancestries and expression intensities [43].

A total of 700 expressive face images (1 intensity × 6 expressions × 100 subjects) and 100 neutral face images (each of which was of one subject) [43] were used.

- JAFFE data set: This contains 10 female subjects and 213 facial expression images [55]. The number of images corresponding to each of the seven categories of expression (neutrality, happiness, sadness, surprise, anger, disgust, and fear) was almost always the same. Each actor repeated the same expression several times (i.e., two, three, or four times).

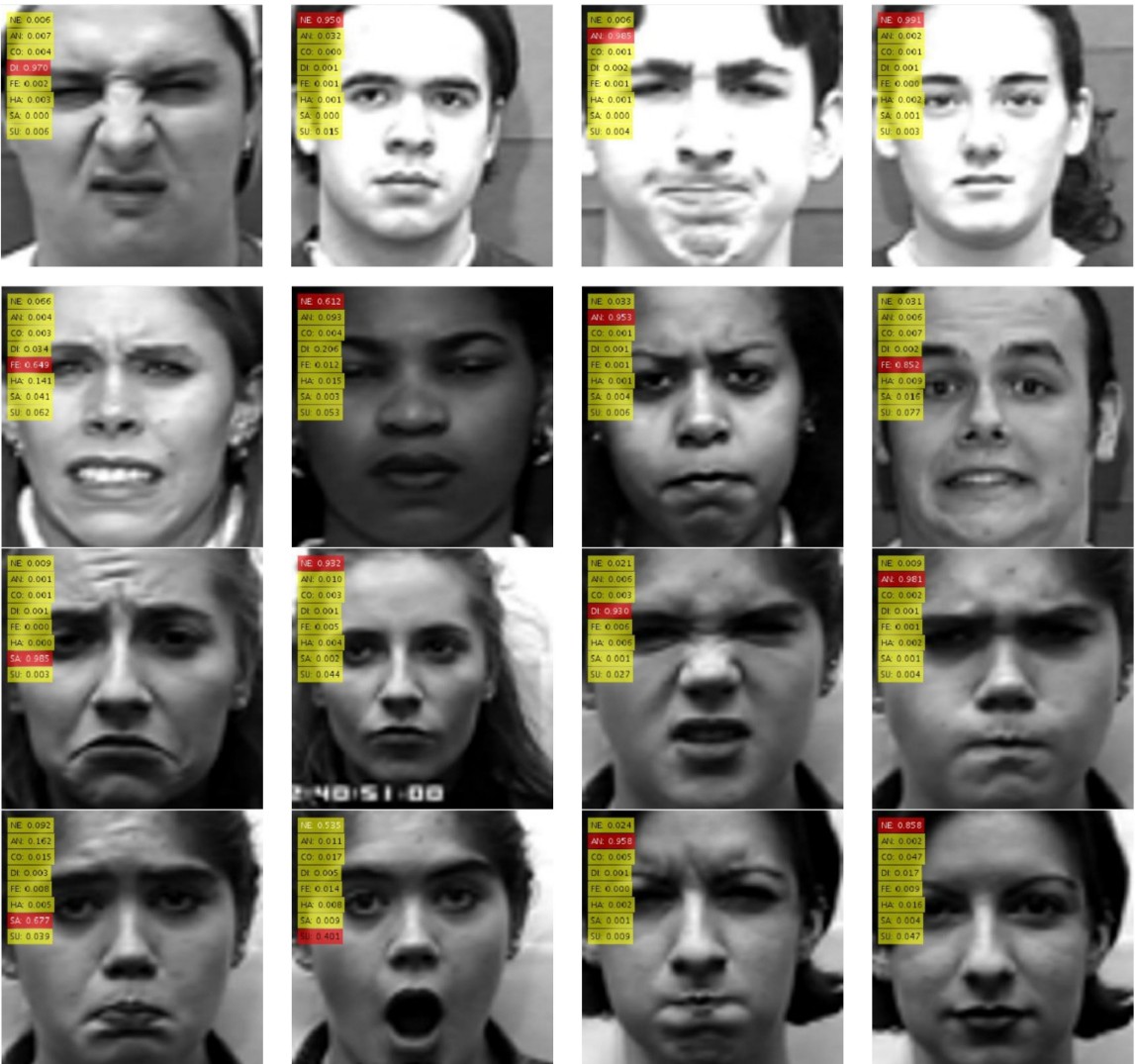

**Figure 3.** Classification results. The two top rows show different users, and the two bottom rows show each user with two different expressions. We denote the expression marked by the classifier system in red. Original images extracted from BU-3DFE [42,43].

### 4.2. Protocol

In this work, eight feature extraction methods—GW, LBP, HOG, LPQ, RAW, GEO, VGG, and VGGF—were used, according to the methodology proposed. These features have been used widely in the literature for FER [17–19,27,30,39,41,57,58]. We generated 247 possible scenarios, which were all combinations of the selected feature extraction method. We denote the classification schemes as follows: G**W**+SR with W, L**B**P+SR with B, **H**oG+SR with H, L**P**Q+SR with P, **R**AW+SR with R, **G**EO+SR with G, **V**GG+SR with V, and VGG**F**+SR with F.

Next, for example, a possible combination scenario might be denoted as W/B/H, which corresponds to G**W**/L**B**P/**H**OG+SR. We tested 14 combination rules for each scenario: five soft-level rules—product rule (RP), sum rule (RS), max rule (RMX), min rule (RMI), and median rule (RMD) [20]; three hard-level rules—weighted majority vote rule (WMV), recall rule (REC), and naïve Bayes rule (NB) [59]; and six trainable methods—fuzzy integral (FI) [51], linear opinion pools (LOP) [52], Bayes model (MB), SVM linear kernel model (ML), and SVM polynomial kernel model (MP). The results of these combinations were also compared to those of the individual methods.

The parameters for each of the methods were selected according to those obtained using state-of-the-art methods. The GW representation was obtained using five scales and eight orientations, in order to construct a set of Gabor filter banks for $25 \times 25$ and $k = 7$ neighborhoods [60]; the image resolution was $256 \times 256$ pixels. For the extraction of LBP features, as used by Huang et al. [38] and Ying et al. [17], we adopted a uniform LBP operation, with parameters of $P = 8$ and $R = 2$ in images sized $256 \times 256$ pixels. The histogram was extracted for each patch sized $25 \times 25$. For HOG, the bin number was set to 9, the cell size was $16 \times 16$ pixels, and a block size of $2 \times 2$ was adopted for each selected landmark point [19]. For the extraction of the LPQ [28], we set $M = 5$ and $a = 1/5$. The histograms were extracted from each selected landmark point for those patches sized $25 \times 25$. In the case of VGGF, as a feature, we selected the output of the ReLU layer 33; while, in the case of VGG, we selected the output of the ReLU layer 34.

*4.3. Metrics*

For this analysis, different metrics were used. Accuracy was calculated as the average number of successes, divided by the total number of observations (in this case, each face was considered an observation). The precision, recall, F1-score, and confusion matrix were also used for an analysis of the effectiveness of the system. Demšar [61] recommended the Friedman test, followed by the pairwise Nemenyi test, to compare multiple pieces of data. The Friedman test is a non-parametric alternative to the ANOVA test. The null hypothesis of the test $H_0$ is that all classifier models are equivalent. In our work, the Friedman test was used to identify the best classification scheme between different combination rules and all feature extraction methods. Similar to previously published works [31,62–64], leave-one-subject-out (LOSO) cross-validation was adopted in the evaluation.

*4.4. Experimental Environment*

One of the most important aspects to consider when evaluating the use of multiple classifier systems is that of time. A critical component of the system is the feature extraction method ($Fs_i$) and its representation by SRC. As the number of methods increases, the amount of time also increases. It was expected that method $Fs_1$ + SRC has time $t_1$, method $Fs_2$ + SRC has time $t_2$, method $Fs_n$+ SRC has time $t_n$, and the system time is $t = \sum t_i$. However, the system components are independent and depend only on the input image (i.e., there are no interdependencies); therefore, if the number of methods in the pool is not large, each one can be assigned to a processing unit. In this case, the mean time could be shown as $t = max(t_i) + c$, where $c$ is a constant. At present, such solutions are viable and easily accessible.

The hardware used to carry out the computation was a desktop computer with the following specifications: CPU: Intel i9-9900KF (16) @ 5.000 GHz; GPU: NVIDIA GeForce RTX 2060; memory: 7003 MiB/32,035 MiB; OS: Ubuntu 20.04.4; LTS: x86_64. With this configuration, the total computing time was 12.3 h (including feature extraction, sparse representation, training, and metrics).

## 5. Results and Discussion

In this section, we present the results and discuss the influence of the combination rule, the differences in the use of individual and multiple classification methods, and the contribution of each feature.

### 5.1. Statistical Analysis of the Combination Rule

Tables 3–5 show the accuracy values for the 14 combination rules (columns) and the best combinations of classifiers in different subspaces (rows) for the CK+ data set, BU-3DFE data set, and JAFFE data set, respectively. Regarding the CK+ data set (Table 3), an accuracy of more than 0.985 was achieved; in the BU-3DFE data set (Table 4), we achieved an accuracy of more than 0.821; and for JAFFE (Table 5), the accuracy was 0.776 (in all three cases, more than 20 combinations were selected). The best accuracy for each combination is underlined. The last row of tables shows the average accuracy across the 247 combinations. With the large span of classification accuracy values, it is unlikely that these values will be commensurable; however, although the average values across the feature combination do not serve as a valid performance metric, they give a rough indication of the achievements of the combinations. The tables show that the trainable combination rules of MP and ML presented the best results in most cases. The JAFFE data set had the fewest images (213 images), thus limiting the training phase of the trainable combiners. In this case, the best option seemed to be the use of non-trainable combiners.

**Table 3.** Accuracy selection was greater than 0.985 for the 14-rule combination methods and 247 combinations of eight subspaces in the CK+ data set. CSC: a combination of soft classifiers; CHC: a combination of hard classifiers; TC: trainable classifiers. Best classifier result is underlined. Best feature/classifier is in bold type.

| Features | CSC | | | | | | CHC | | | | TC | | | |
|---|---|---|---|---|---|---|---|---|---|---|---|---|---|---|
| | RP | RS | RMX | RMI | RMD | RM | WMV | REC | NB | LOP | FI | MB | MP | ML |
| R/B/P/V | 0.986 | 0.986 | 0.950 | 0.913 | 0.986 | 0.975 | 0.972 | 0.969 | 0.958 | 0.978 | 0.950 | 0.978 | 0.975 | 0.975 |
| R/P/V/F | 0.961 | 0.961 | 0.936 | 0.908 | 0.961 | 0.944 | 0.947 | 0.958 | 0.939 | 0.955 | 0.936 | 0.980 | 0.989 | 0.983 |
| R/W/B/P/V | 0.975 | 0.975 | 0.947 | 0.930 | 0.975 | 0.972 | 0.964 | 0.950 | 0.950 | 0.975 | 0.947 | 0.986 | 0.978 | 0.978 |
| R/W/B/V/F | 0.969 | 0.969 | 0.947 | 0.933 | 0.969 | 0.964 | 0.964 | 0.961 | 0.955 | 0.961 | 0.947 | 0.986 | 0.983 | 0.980 |
| R/W/H/P/F | 0.969 | 0.969 | 0.947 | 0.941 | 0.969 | 0.958 | 0.955 | 0.961 | 0.947 | 0.964 | 0.947 | 0.989 | 0.983 | 0.983 |
| R/W/P/G/F | 0.947 | 0.947 | 0.902 | 0.894 | 0.947 | 0.953 | 0.953 | 0.961 | 0.941 | 0.925 | 0.902 | 0.972 | 0.986 | 0.986 |
| R/B/H/V/F | 0.969 | 0.969 | 0.953 | 0.933 | 0.969 | 0.966 | 0.966 | 0.953 | 0.950 | 0.966 | 0.953 | 0.975 | 0.980 | 0.986 |
| R/H/P/V/F | 0.964 | 0.964 | 0.944 | 0.916 | 0.964 | 0.950 | 0.955 | 0.953 | 0.927 | 0.953 | 0.944 | 0.983 | 0.986 | 0.986 |
| R/P/G/V/F | 0.939 | 0.939 | 0.883 | 0.880 | 0.939 | 0.933 | 0.941 | 0.961 | 0.908 | 0.930 | 0.883 | 0.975 | 0.989 | 0.980 |
| W/B/H/P/V | 0.975 | 0.975 | 0.955 | 0.927 | 0.975 | 0.964 | 0.964 | 0.953 | 0.950 | 0.964 | 0.955 | 0.986 | 0.983 | 0.983 |
| W/B/H/V/F | 0.969 | 0.969 | 0.955 | 0.930 | 0.969 | 0.961 | 0.958 | 0.947 | 0.936 | 0.958 | 0.955 | 0.978 | 0.986 | 0.983 |
| R/W/B/H/P/V | 0.978 | 0.978 | 0.947 | 0.933 | 0.978 | 0.975 | 0.972 | 0.955 | 0.955 | 0.972 | 0.947 | 0.989 | 0.986 | 0.983 |
| R/W/B/H/P/F | 0.975 | 0.975 | 0.953 | 0.944 | 0.975 | 0.978 | 0.969 | 0.953 | 0.964 | 0.975 | 0.953 | 0.986 | 0.983 | 0.983 |
| R/W/B/H/V/F | 0.975 | 0.975 | 0.947 | 0.939 | 0.975 | 0.964 | 0.961 | 0.950 | 0.950 | 0.961 | 0.947 | 0.989 | 0.980 | 0.986 |
| R/W/B/P/G/V | 0.958 | 0.958 | 0.908 | 0.888 | 0.958 | 0.972 | 0.966 | 0.950 | 0.941 | 0.941 | 0.908 | 0.986 | 0.983 | 0.983 |
| R/W/B/P/V/F | 0.975 | 0.975 | 0.947 | 0.927 | 0.975 | 0.975 | 0.969 | 0.961 | 0.961 | 0.961 | 0.947 | 0.989 | 0.980 | 0.980 |
| R/W/H/P/G/V | 0.958 | 0.958 | 0.897 | 0.885 | 0.958 | 0.955 | 0.953 | 0.950 | 0.919 | 0.933 | 0.897 | 0.980 | 0.980 | 0.986 |
| R/W/H/P/G/F | 0.955 | 0.955 | 0.905 | 0.894 | 0.955 | 0.958 | 0.955 | 0.953 | 0.941 | 0.933 | 0.905 | 0.978 | 0.986 | 0.986 |
| R/W/H/P/V/F | 0.966 | 0.966 | 0.936 | 0.927 | 0.966 | 0.966 | 0.955 | 0.953 | 0.939 | 0.955 | 0.936 | 0.986 | **0.992** | 0.989 |
| R/B/H/P/V/F | 0.972 | 0.972 | 0.953 | 0.925 | 0.972 | 0.975 | 0.969 | 0.955 | 0.950 | 0.966 | 0.953 | 0.989 | 0.986 | 0.986 |
| R/B/H/G/V/F | 0.961 | 0.961 | 0.899 | 0.888 | 0.961 | 0.964 | 0.964 | 0.953 | 0.947 | 0.933 | 0.899 | 0.975 | 0.986 | 0.986 |
| R/H/P/G/V/F | 0.944 | 0.947 | 0.894 | 0.883 | 0.947 | 0.958 | 0.955 | 0.947 | 0.902 | 0.927 | 0.894 | 0.978 | 0.989 | 0.986 |
| W/B/H/P/V/F | 0.972 | 0.972 | 0.955 | 0.925 | 0.972 | 0.972 | 0.966 | 0.955 | 0.939 | 0.953 | 0.955 | 0.983 | 0.980 | 0.986 |
| R/W/B/H/P/G/F | 0.966 | 0.964 | 0.911 | 0.894 | 0.964 | 0.966 | 0.961 | 0.953 | 0.964 | 0.925 | 0.911 | 0.986 | 0.983 | 0.983 |
| R/W/B/H/P/V/F | 0.972 | 0.972 | 0.947 | 0.933 | 0.972 | 0.966 | 0.969 | 0.955 | 0.955 | 0.958 | 0.947 | 0.989 | 0.980 | 0.983 |
| R/W/B/P/G/V/F | 0.958 | 0.958 | 0.908 | 0.888 | 0.958 | 0.966 | 0.958 | 0.955 | 0.958 | 0.933 | 0.908 | 0.986 | 0.983 | 0.983 |
| R/W/H/P/G/V/F | 0.950 | 0.950 | 0.897 | 0.885 | 0.950 | 0.953 | 0.947 | 0.953 | 0.927 | 0.930 | 0.897 | 0.980 | 0.989 | 0.989 |
| W/B/H/P/G/V/F | 0.955 | 0.955 | 0.891 | 0.880 | 0.955 | 0.961 | 0.961 | 0.953 | 0.939 | 0.930 | 0.891 | 0.983 | 0.986 | 0.986 |
| Average | 0.946 | 0.946 | 0.910 | 0.902 | 0.946 | 0.941 | 0.946 | 0.948 | 0.929 | 0.940 | 0.910 | 0.966 | 0.971 | 0.971 |

To determine the best choice for CK+ and BU-3DFE, we calculated the ranks of the combiners. For example, for the combined features R/W/H/P/V/F in the CK+ dataset, the order by rank was as follows: MP (the best), ML, MB, RP, RS, RMD, MV, WMV, LOP, REC, NB, RMX, FI, and RMI (the worst). In case of a tie, the ranks were shared. The average ranks across the combination of features in CK+ (BU-3DFE) were as follows: RP 6.709 (5.830), RS 6.749 (5.873), RMX 11.699 (12.694), RMI 12.956 (12.484), RMD 6.749 (5.873), MV 7.820 (9.982), WMV 7.524 (7.401), REC 7.512 (6.328), NB 10.757 (9.423), LOP 8.124 (7.294),

FI 11.682 (12.611), MB 2.767 (3.792), MP 1.984 (2.678), and ML 1.970 (2.739). These results indicate that MP, ML, and MB were the best combiners in both cases.

**Table 4.** Accuracy selection greater than 0.821 for 14-rule combination methods and 247 combinations of eight subspaces in the BU-3DFE data set. CSC: a combination of soft classifiers; CHC: a combination of hard classifiers; TC: trainable classifiers. Best classifier result is underlined. Best feature/classifier is in bold type.

| Features | CSC | | | | | | CHC | | | | TC | | | |
|---|---|---|---|---|---|---|---|---|---|---|---|---|---|---|
| | RP | RS | RMX | RMI | RMD | RM | WMV | REC | NB | LOP | FI | MB | MP | ML |
| R/W/P/G | 0.774 | 0.778 | 0.724 | 0.697 | 0.778 | 0.772 | 0.783 | 0.781 | 0.771 | 0.771 | 0.724 | 0.807 | 0.824 | <u>0.826</u> |
| R/W/B/P/G | 0.807 | 0.807 | 0.741 | 0.709 | 0.807 | 0.781 | 0.798 | 0.802 | 0.779 | 0.788 | 0.741 | 0.807 | <u>0.828</u> | 0.826 |
| R/W/B/P/V | 0.802 | 0.800 | 0.752 | 0.767 | 0.800 | 0.790 | 0.797 | 0.795 | 0.778 | 0.797 | 0.752 | <u>0.822</u> | 0.816 | 0.817 |
| R/W/H/P/G | 0.788 | 0.791 | 0.729 | 0.714 | 0.791 | 0.805 | 0.805 | 0.809 | 0.779 | 0.784 | 0.729 | 0.809 | <u>0.826</u> | 0.824 |
| R/H/P/G/V | 0.784 | 0.784 | 0.724 | 0.714 | 0.784 | 0.779 | 0.791 | 0.790 | 0.778 | 0.776 | 0.724 | 0.803 | <u>0.824</u> | 0.819 |
| W/H/P/G/V | 0.778 | 0.778 | 0.719 | 0.698 | 0.778 | 0.779 | 0.788 | 0.800 | 0.784 | 0.781 | 0.719 | 0.795 | <u>0.822</u> | 0.819 |
| W/H/P/V/F | 0.795 | 0.798 | 0.734 | 0.748 | 0.798 | 0.798 | 0.793 | 0.797 | 0.781 | 0.783 | 0.734 | <u>0.822</u> | 0.814 | 0.810 |
| R/W/B/H/P/G | 0.809 | 0.807 | 0.743 | 0.712 | 0.807 | 0.802 | 0.800 | 0.819 | 0.783 | 0.798 | 0.743 | 0.819 | <u>0.828</u> | 0.822 |
| R/W/B/H/P/F | 0.817 | 0.816 | 0.745 | 0.752 | 0.816 | 0.810 | 0.807 | 0.817 | 0.795 | 0.809 | 0.745 | 0.821 | 0.822 | <u>0.824</u> |
| R/W/B/P/G/V | 0.798 | 0.800 | 0.741 | 0.721 | 0.800 | 0.788 | 0.795 | 0.795 | 0.783 | 0.783 | 0.741 | 0.814 | <u>0.826</u> | 0.826 |
| R/W/H/P/G/V | 0.800 | 0.800 | 0.728 | 0.721 | 0.800 | 0.798 | 0.809 | 0.814 | 0.791 | 0.790 | 0.728 | 0.816 | <u>0.826</u> | 0.824 |
| R/W/H/P/G/F | 0.803 | 0.803 | 0.728 | 0.722 | 0.803 | 0.798 | 0.807 | <u>0.828</u> | 0.784 | 0.793 | 0.728 | 0.816 | 0.812 | 0.807 |
| R/B/H/P/V/F | 0.807 | 0.807 | 0.764 | 0.767 | 0.807 | 0.803 | 0.810 | 0.814 | 0.802 | 0.800 | 0.764 | <u>0.822</u> | 0.810 | 0.810 |
| R/H/P/G/V/F | 0.810 | 0.809 | 0.722 | 0.724 | 0.809 | 0.786 | 0.803 | 0.812 | 0.779 | 0.776 | 0.722 | 0.817 | 0.822 | <u>0.824</u> |
| W/B/H/P/G/V | 0.800 | 0.798 | 0.741 | 0.707 | 0.798 | 0.795 | 0.795 | 0.807 | 0.790 | 0.784 | 0.741 | 0.809 | <u>0.822</u> | 0.821 |
| W/B/H/P/G/F | 0.803 | 0.803 | 0.726 | 0.710 | 0.803 | 0.805 | 0.807 | 0.819 | 0.790 | 0.790 | 0.726 | 0.812 | <u>0.822</u> | 0.822 |
| W/B/H/P/V/F | 0.805 | 0.803 | 0.750 | 0.759 | 0.803 | 0.805 | 0.809 | 0.809 | 0.790 | 0.786 | 0.750 | <u>0.828</u> | 0.814 | 0.814 |
| R/W/B/H/P/G/V | 0.802 | 0.800 | 0.743 | 0.722 | 0.800 | 0.791 | 0.802 | 0.816 | 0.790 | 0.786 | 0.790 | <u>0.821</u> | 0.822 | 0.819 |
| R/W/B/H/P/G/F | 0.821 | 0.821 | 0.741 | 0.722 | 0.821 | 0.816 | 0.819 | 0.812 | 0.795 | 0.798 | 0.791 | **<u>0.828</u>** | 0.826 | 0.822 |
| R/W/B/H/P/V/F | 0.802 | 0.803 | 0.750 | 0.760 | 0.803 | 0.812 | 0.816 | 0.817 | 0.798 | 0.795 | 0.791 | <u>0.824</u> | 0.816 | 0.816 |
| Average | 0.779 | 0.779 | 0.727 | 0.727 | 0.779 | 0.762 | 0.774 | 0.777 | 0.766 | 0.7740 | 0.729 | 0.789 | <u>0.794</u> | 0.794 |

**Table 5.** Accuracy selection greater than 0.776 for 14-rule combination methods and 247 combinations of eight subspaces in the JAFFE data set. CSC: a combination of soft classifiers; CHC: a combination of hard classifiers; TC: trainable classifiers. Best classifier result is underlined. Best feature/classifier is in bold type.

| Features | CSC | | | | | | CHC | | | | TC | | | |
|---|---|---|---|---|---|---|---|---|---|---|---|---|---|---|
| | RP | RS | RMX | RMI | RMD | RM | WMV | REC | NB | LOP | FI | MB | MP | ML |
| R/B/H | 0.746 | 0.746 | 0.726 | 0.706 | 0.746 | 0.701 | 0.711 | 0.706 | 0.692 | <u>0.761</u> | 0.726 | 0.751 | 0.726 | 0.741 |
| B/H/P | 0.746 | 0.746 | 0.706 | 0.667 | 0.746 | 0.726 | 0.706 | 0.692 | 0.701 | <u>0.766</u> | 0.706 | 0.716 | 0.736 | 0.736 |
| R/W/H/G | 0.751 | 0.746 | 0.647 | 0.622 | 0.746 | 0.721 | 0.731 | 0.736 | 0.597 | 0.731 | 0.647 | <u>0.761</u> | 0.736 | 0.736 |
| R/H/P/G | <u>0.771</u> | 0.766 | 0.672 | 0.612 | 0.766 | 0.701 | 0.721 | 0.711 | 0.597 | 0.716 | 0.672 | 0.731 | 0.736 | 0.736 |
| R/H/P/F | 0.716 | 0.716 | 0.697 | 0.687 | 0.716 | 0.726 | 0.711 | 0.697 | 0.627 | 0.736 | 0.697 | 0.746 | **<u>0.766</u>** | 0.761 |
| W/B/H/P | 0.721 | 0.721 | 0.682 | 0.682 | 0.721 | 0.687 | 0.706 | 0.751 | 0.647 | 0.751 | 0.682 | 0.721 | <u>0.761</u> | 0.756 |
| W/H/P/F | 0.746 | 0.746 | 0.647 | 0.677 | 0.746 | 0.726 | 0.706 | 0.726 | 0.642 | 0.736 | 0.647 | 0.721 | 0.761 | <u>0.776</u> |
| R/W/H/P/G | 0.751 | 0.751 | 0.647 | 0.622 | 0.751 | 0.716 | 0.731 | 0.736 | 0.612 | 0.741 | 0.647 | <u>0.761</u> | 0.746 | 0.746 |
| R/W/H/G/F | <u>0.766</u> | 0.761 | 0.642 | 0.617 | 0.761 | 0.716 | 0.726 | 0.726 | 0.562 | 0.721 | 0.642 | 0.741 | 0.692 | 0.687 |
| R/B/H/G/F | <u>0.761</u> | 0.756 | 0.677 | 0.617 | 0.756 | 0.731 | 0.716 | 0.701 | 0.582 | 0.711 | 0.677 | 0.716 | 0.721 | 0.726 |
| R/H/P/G/F | <u>0.766</u> | 0.766 | 0.667 | 0.617 | 0.766 | 0.721 | 0.721 | 0.706 | 0.587 | 0.701 | 0.667 | 0.736 | 0.751 | 0.751 |
| W/H/P/V/F | 0.741 | 0.746 | 0.652 | 0.637 | 0.746 | 0.716 | 0.731 | 0.716 | 0.587 | 0.726 | 0.652 | 0.716 | <u>0.766</u> | 0.756 |
| R/W/B/H/P/F | 0.741 | 0.741 | 0.692 | 0.682 | 0.741 | 0.726 | 0.721 | 0.726 | 0.637 | 0.731 | 0.692 | 0.731 | 0.746 | <u>0.761</u> |
| R/W/B/H/G/V | 0.731 | 0.731 | 0.652 | 0.612 | 0.731 | 0.716 | 0.721 | 0.721 | 0.522 | 0.697 | 0.652 | <u>0.771</u> | 0.726 | 0.731 |
| R/W/B/H/G/F | <u>0.761</u> | 0.761 | 0.657 | 0.622 | 0.761 | 0.726 | 0.731 | 0.721 | 0.587 | 0.716 | 0.657 | 0.746 | 0.716 | 0.726 |
| R/W/B/H/V/F | <u>0.761</u> | 0.756 | 0.692 | 0.637 | 0.756 | 0.716 | 0.731 | 0.731 | 0.572 | 0.721 | 0.692 | 0.741 | 0.697 | 0.692 |
| R/W/H/P/G/F | <u>0.766</u> | 0.766 | 0.642 | 0.622 | 0.766 | 0.726 | 0.741 | 0.736 | 0.582 | 0.716 | 0.642 | 0.741 | 0.736 | 0.736 |
| R/B/H/P/G/F | 0.766 | <u>0.771</u> | 0.677 | 0.622 | 0.771 | 0.741 | 0.726 | 0.706 | 0.607 | 0.706 | 0.677 | 0.731 | 0.746 | 0.736 |
| R/B/H/P/V/F | <u>0.761</u> | 0.761 | 0.687 | 0.647 | 0.761 | 0.746 | 0.751 | 0.731 | 0.602 | 0.746 | 0.687 | 0.736 | 0.721 | 0.711 |
| R/W/B/H/P/G/F | <u>0.771</u> | 0.771 | 0.657 | 0.627 | 0.771 | 0.731 | 0.731 | 0.726 | 0.597 | 0.721 | 0.756 | 0.746 | 0.756 | 0.756 |
| R/W/B/H/P/V/F | <u>0.761</u> | 0.761 | 0.692 | 0.647 | 0.761 | 0.726 | 0.726 | 0.731 | 0.592 | 0.721 | 0.726 | 0.731 | 0.721 | 0.721 |
| R/B/H/P/G/V/F | <u>0.761</u> | 0.761 | 0.657 | 0.597 | 0.761 | 0.746 | 0.751 | 0.736 | 0.562 | 0.711 | 0.741 | 0.726 | 0.736 | 0.731 |
| Average | 0.700 | 0.700 | 0.640 | 0.604 | 0.700 | 0.685 | 0.700 | 0.697 | 0.586 | 0.691 | 0.643 | 0.693 | 0.685 | 0.686 |

The Friedman non-parametric test was executed on the ranks, followed by a multiple comparisons test. The Friedman test score was 2326.60 (2249.70), giving a *p*-value of approximately 0 (0), indicating significant differences among the ranks for the CK+ (BU-3DFE) data set. The Nemenyi post hoc test and Bonferroni–Dunn post hoc test were applied, in order to determine those methods that had significant differences.

The result of the Nemenyi post hoc test (two-tailed test) showed that there were significant differences between the MB, MP, and ML methods and all of the other methods, with a significance level of $\alpha < 0.05$. For MB, MP, and ML, we applied the Bonferroni–Dunn post hoc test (one-tailed test) to strengthen the power of the test hypotheses. At a significance level of 0.05, the Bonferroni–Dunn post hoc test did not show significant differences between the MB, MP, and ML methods. Therefore, we can conclude that, in general, these methods had similar behavior in this case. For the case of the BU-3DFE data set, similar results were obtained. The proposed methods (MB, MP, and ML) were significantly superior to the others for combination rules in FER problems; however, when very little training data sets are available, the use of non-trainable combination rules is suggested.

Tables 6–8 show the accuracy, precision, recall, and F1-score measurements of the best schemes for each individual and combination rule in the CK+ data set, BU-3DFE data set, and JAFFE data set, respectively. In the case of the CK+ data set (Table 6), the R/W/H/P/G/F+MP scheme presented an accuracy of 0.992 and also obtained the best results in terms of precision, recall, and F1-score, with values of 0.991, 0.985, and 0.999, respectively. In the case of the BU-3DFE data set (Table 7), the R/W/B/P/G+MP scheme presented an 0.828 accuracy, with the best values of precision, recall, and F1-score also being obtained by this scheme (0.893, 0.833, and 0.965, respectively). In both cases, the MP combination rule achieved the best results. In the case of the JAFFE data set (Table 8), the W/H/P/F+ML scheme obtained an accuracy of 0.828; however, the R/H/P/G+RP scheme showed the best results for precision, recall, and F1 (0.871, 0.809, and 0.959, respectively). This was because, as mentioned earlier, the trainable combination rules are inadequate when there are insufficient data in the training data set.

In summary, our results exceeded the results of the state-of-the-art for these types of features. In all cases, the combination rules improved upon the accuracy of the single methods. Figure 4 shows that the classification error of the individual methods was greater than that of the combination schemes.

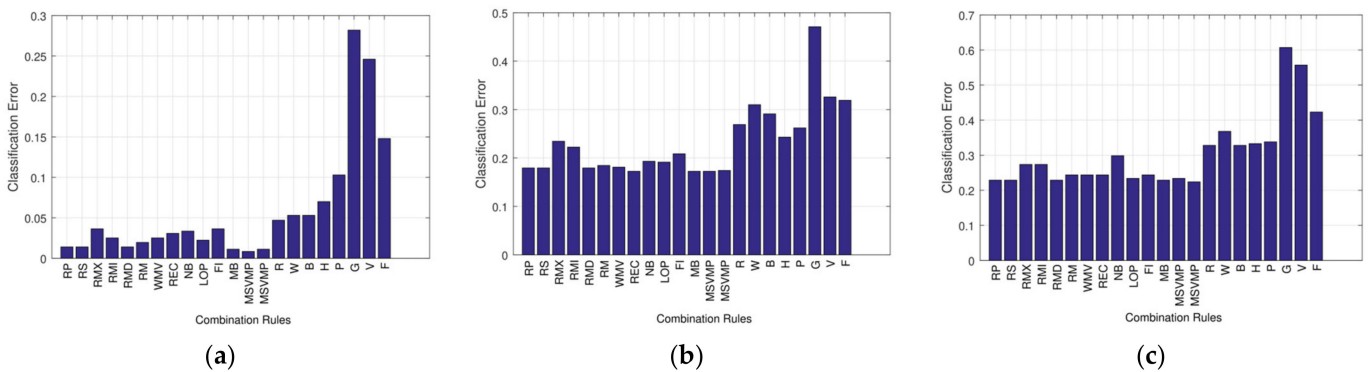

**Figure 4.** Classification error of the best schemes for each combination rule and individual schemes: (**a**) CK+ data set; (**b**) BU-3DFE data set; (**c**) JAFFE data set.

**Table 6.** The best schemes for each combination rule ($e \cdot 1000^{-1}$) for the CK+ data set. Acc: accuracy; Prec: precision; Rec: recall. Best classifier result is underlined.

| | Combined Schemes | | | | | | | | | | | | | | Individual Schemes | | | | | | | |
|---|---|---|---|---|---|---|---|---|---|---|---|---|---|---|---|---|---|---|---|---|---|---|
| | RP (R/B/P/V) | RS (R/B/P/V) | RMX (R/B) | RMI (W/B) | RMD (R/B/P/V) | RM (R/W/B/P) | WMV (R/B/H/F) | REC (R/B/P/V) | NB (R/B/H/F) | LOP (R/W/B) | FI (R/B) | MB (R/W/H/P/F) | MSVMP (R/W/H/P/G/F) | MSVML (R/W/H/P/G/F) | R | B | H | W | P | G | V | F |
| **Acc.** | 0.986 | 0.986 | 0.964 | 0.975 | 0.986 | 0.980 | 0.975 | 0.969 | 0.966 | 0.978 | 0.964 | 0.989 | <u>0.992</u> | 0.989 | 0.953 | 0.947 | 0.930 | 0.947 | 0.897 | 0.718 | 0.754 | 0.852 |
| **F1** | 0.992 | 0.992 | 0.968 | 0.982 | 0.992 | 0.987 | 0.981 | 0.975 | 0.973 | 0.982 | 0.968 | 0.989 | <u>0.991</u> | 0.989 | 0.961 | 0.966 | 0.951 | 0.964 | 0.940 | 0.774 | 0.798 | 0.898 |
| **Prec** | 0.985 | 0.985 | 0.945 | 0.968 | 0.985 | 0.977 | 0.967 | 0.957 | 0.953 | 0.968 | 0.945 | 0.980 | <u>0.985</u> | 0.980 | 0.936 | 0.942 | 0.918 | 0.938 | 0.900 | 0.674 | 0.699 | 0.835 |
| **Rec** | 0.998 | 0.998 | 0.994 | 0.996 | 0.998 | 0.997 | 0.996 | 0.995 | 0.995 | 0.997 | 0.994 | 0.998 | <u>0.999</u> | 0.998 | 0.993 | 0.992 | 0.990 | 0.992 | 0.985 | 0.936 | 0.948 | 0.976 |

**Table 7.** The best schemes for each combination rule ($e \cdot 1000^{-1}$) for the BU-3DFE data set. Acc: accuracy; Prec: precision; Rec: recall. Best classifier result is underlined.

| | Combined Schemes | | | | | | | | | | | | | | Individual Schemes | | | | | | | |
|---|---|---|---|---|---|---|---|---|---|---|---|---|---|---|---|---|---|---|---|---|---|---|
| | RP (R/B/H/P/G/F) | RS (R/B/H/P/G/F) | RMX (H/P) | RMI (R/H/V) | RMD (R/B/H/P/G/F) | RM (R/W/B/H/P/G/F) | WMV (R/W/B/H/P/G/F) | REC (R/W/H/P/G/F) | NB (R/B/H/P/F) | LOP (R/W/B/H/P/F) | FI (R/W/B/H/P/G/F) | MB (W/B/H/P/V/F) | MSVMP (R/W/B/P/G) | MSVML (R/W/P/G) | R | B | H | W | P | G | V | F |
| **Acc.** | 0.821 | 0.821 | 0.766 | 0.778 | 0.821 | 0.816 | 0.819 | 0.828 | 0.807 | 0.809 | 0.791 | 0.828 | <u>0.828</u> | 0.826 | 0.731 | 0.709 | 0.757 | 0.690 | 0.738 | 0.529 | 0.674 | 0.681 |
| **F1** | 0.891 | 0.890 | 0.849 | 0.858 | 0.890 | 0.890 | 0.890 | 0.894 | 0.878 | 0.883 | 0.874 | 0.892 | <u>0.893</u> | 0.892 | 0.833 | 0.810 | 0.842 | 0.799 | 0.840 | 0.654 | 0.782 | 0.791 |
| **Prec** | 0.829 | 0.828 | 0.770 | 0.780 | 0.828 | 0.830 | 0.830 | 0.835 | 0.812 | 0.819 | 0.807 | 0.830 | <u>0.833</u> | 0.830 | 0.753 | 0.720 | 0.760 | 0.706 | 0.759 | 0.537 | 0.687 | 0.697 |
| **Rec** | 0.965 | 0.965 | 0.950 | 0.954 | 0.965 | 0.964 | 0.964 | 0.965 | 0.960 | 0.962 | 0.958 | 0.965 | 0.965 | <u>0.966</u> | 0.941 | 0.933 | 0.947 | 0.929 | 0.947 | 0.857 | 0.919 | 0.924 |

**Table 8.** The best schemes for each combination rule ($e \cdot 1000^{-1}$) for the JAFFE data set. Acc: accuracy; Prec: precision; Rec: recall. Best classifier result is underlined.

| | Combined Schemes | | | | | | | | | | | | | | Individual Schemes | | | | | | | |
| | RP (R/H/P/G) | RS (R/B/H/P/G/F) | RMX (R/B) | RMI (R/H) | RMD (R/B/H/P/G/F) | RM (R/B/H/P/V) | WMV (R/B/H/G/V) | REC (R/W/H/P/G/V) | NB (B/H/P) | LOP (B/H/P) | FI (R/W/B/H/P/G/F) | MB (R/W/B/H/G/V) | MSVMP (R/H/P/F) | MSVML (W/H/P/F) | R | B | H | W | P | G | V | F |
|---|---|---|---|---|---|---|---|---|---|---|---|---|---|---|---|---|---|---|---|---|---|---|
| **Acc.** | 0.771 | 0.771 | 0.726 | 0.726 | 0.771 | 0.756 | 0.756 | 0.756 | 0.701 | 0.766 | 0.756 | 0.771 | 0.766 | <u>0.776</u> | 0.672 | 0.672 | 0.667 | 0.632 | 0.662 | 0.393 | 0.443 | 0.577 |
| **F1** | <u>0.871</u> | 0.870 | 0.838 | 0.830 | 0.870 | 0.867 | 0.850 | 0.848 | 0.803 | 0.859 | 0.860 | 0.856 | 0.844 | 0.859 | 0.787 | 0.785 | 0.787 | 0.733 | 0.793 | 0.541 | 0.578 | 0.701 |
| **Prec** | <u>0.809</u> | 0.807 | 0.764 | 0.750 | 0.807 | 0.804 | 0.776 | 0.768 | 0.707 | 0.788 | 0.794 | 0.779 | 0.762 | 0.784 | 0.689 | 0.694 | 0.695 | 0.629 | 0.699 | 0.416 | 0.462 | 0.583 |
| **Rec** | <u>0.959</u> | 0.958 | 0.944 | 0.942 | 0.958 | 0.957 | 0.950 | 0.951 | 0.934 | 0.955 | 0.953 | 0.954 | 0.950 | 0.956 | 0.927 | 0.924 | 0.926 | 0.898 | 0.932 | 0.804 | 0.809 | 0.889 |

### 5.2. Multiple vs. Individual Classification Methods

For the experiments, the BU-3DFE and JAFFE data sets were used to construct the training and test sets, respectively. The training and test sets contained six expression classes—anger, disgust, fear, happiness, sadness, and surprise—which were common in both data sets. The experimental results presented in Table 9 show that the fusion method R/W/B/P/G+MP achieved an accuracy of 0.54 and an F1-score of 0.721, higher than those of the other methods used for comparison. Ramis et al. [65] presented a method to combine multiple datasets and conducted an exhaustive evaluation of a proposed system based on a CNN that analyzed and compared performance using single- and cross-dataset approaches with other architectures. They reported that they trained with BU-4FDE and tested with JAFFE, obtaining an accuracy of 0.4317. In our current proposal, we study the most important features by applying different feature extraction methods for facial expression representation, transforming each obtained representation into a sparse representation (SR) domain, and training combination models to classify signals, using the extended Cohn–Kanade (CK+), BU-3DFE, and JAFFE data sets for validation. In general, combining the feature methods increased the system performance. This supports the theory that trainable fusion methods can find experts for each subspace of features and efficiently combine them. In all cases, the combination rules were better at identification than the individual methods. However, the values of the metrics were lower than in the case in which face images from a single data set were used for both the training set and test set. This performance degradation can mainly be attributed to the fact that the face images were collected under two different controlled conditions. For a better generalization ability across image acquisition conditions, it is necessary to collect large training data sets under various image acquisition conditions [66].

**Table 9.** Generalization performance on the two different data sets. The BU-3DFE and JAFFE data sets were used for training and testing, respectively. Best feature results are in bold type.

| Method | Accuracy | F1-Measure |
|---|---|---|
| R/B/H/W/P/G/V/F | 0.523 | 0.726 |
| R/H/P/F | 0.517 | 0.703 |
| R/W/B/H/P/G/F | 0.527 | 0.722 |
| R/W/B/P/G | **0.547** | **0.721** |
| R | 0.433 | 0.635 |
| B | 0.368 | 0.494 |
| H | 0.488 | 0.626 |
| W | 0.353 | 0.594 |
| P | 0.463 | 0.592 |
| G | 0.378 | 0.582 |
| V | 0.189 | 0.332 |
| F | 0.468 | 0.720 |

In summary, our findings confirm the increased accuracy of the proposed combination system, with respect to the individual systems. The best schemes (R/W/H/P/G/F+MP and R/W/B/P/G+MP) combined the best and the worst individual schemes. This increases the diversity of the system, offering the opportunity to select the best representation subspaces for each class.

### 5.3. Analysis of the Influence of the Feature Methods

The visualization of learning models (i.e., the generated subspaces) using classifiers such as SVM and the naïve Bayes method is important. To analyze the contribution of each feature to each class, a logistic regression model was trained on the decision profile. From the weights obtained for each class, a weight map was created. The weight maps obtained for each data set are shown in Figure 5. On the horizontal axis, we can see the weight of the

feature space for each of the classes. On the vertical axis, we can see the importance of each subspace for features over a class.

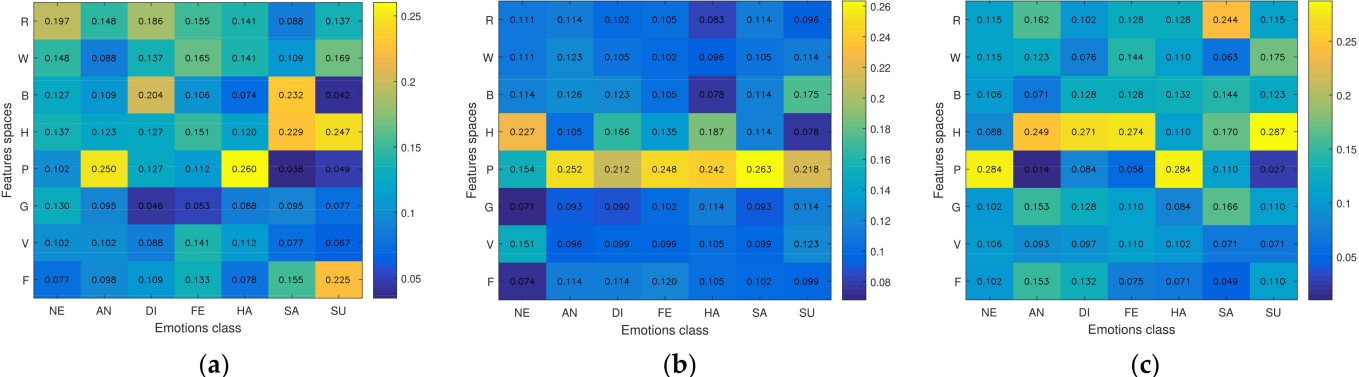

(a)          (b)          (c)

**Figure 5.** Weight map learning for the combination of all features. NE: neutral; AN: anger; DI: disgust; FE: fear; HA: happy; SA: sadness; SU: surprise. (**a**) CK+ data set; (**b**) BU-3DFE data set; and (**c**) JAFFE data set.

As can be seen, the LPQ, HOG, and RAW methods presented high specialization values for some of the classes. For example, for the CK+ data set, LPQ had great decision power over the angry and happy classes, the feature space RAW was most successful in the neutral class, and HOG specialized in the surprise and sad classes. In general, we can observe that the spaces of characteristics do not always specialize in the same classes. This is because there are significant differences between the images in the different data sets.

We calculated the frequencies of the feature methods from Tables 3–5. These frequencies are shown in Figure 6 and Table 10. It can be observed that the LPQ, HOG, and RAW methods were present in more than 80 percent of the selected combinations. This demonstrates that they had significant weight in the classification of some expressions.

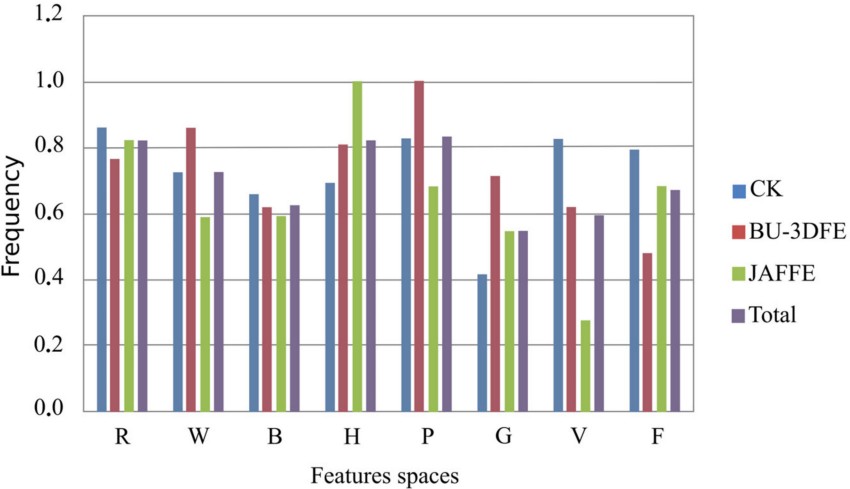

**Figure 6.** Frequency of features in selected combinations (Tables 3–5) for the CK+, BU-3DFE, and JAFFE data sets. RAW: (R), GW: (W), LBP: (B), HOG: (H), LPQ: (P), GEO: (G), VGG: (V), VGGF: (F).

In summary, the LPQ, HOG, and RAW methods were present in most of the selected combinations, and some features were most efficient for particular expressions (e.g., RAW for neutral, LPQ for angry and happy, LBP for disgust, and HOG for surprise).

**Table 10.** Frequency of features in selected combinations for the CK+, BU-3DFE, and JAFFE data sets. RAW: (R), GW: (W), LBP: (B), HOG: (H), LPQ: (P), GEO: (G), VGG: (V), VGGF: (F).

| Feature | CK | BU-3DFE | JAFFE | Total |
|---------|------|---------|-------|-------|
| R | 0.86 | 0.76 | 0.82 | 0.82 |
| W | 0.72 | 0.86 | 0.59 | 0.73 |
| B | 0.66 | 0.62 | 0.59 | 0.40 |
| H | 0.69 | 0.81 | 1.00 | 0.82 |
| P | 0.83 | 1.00 | 0.68 | 0.83 |
| G | 0.42 | 0.71 | 0.54 | 0.55 |
| V | 0.83 | 0.62 | 0.28 | 0.59 |
| F | 0.79 | 0.48 | 0.68 | 0.75 |

## 6. Conclusions

In this work, we studied the different contributions of various combination methods to a facial recognition system. To achieve this, we utilized the concept of sparse representations. The results demonstrated that the combination of classifiers can lead to improved performance when compared to individual classifiers. In particular, the MB, MP, and ML combination methods were significantly superior to other combination rules in facial expression recognition problems.

In all cases, the combination of different features improved the results, compared to when using only one feature. Individually, the best features were **H**oG (in 82% of combinations) and L**P**Q (in 83% of combinations); however, when combining different features, the best selection was **R**AW/**G**W/L**B**P/L**P**Q/**G**EO. Curiously, the best schemes (R/W/H/P/G/F+MP and R/W/B/P/G+MP; accuracy of 0.992 and 0.828, respectively) combined the best and the worst individual schemes. This increases the diversity of the system, providing the opportunity to select the best representation subspace for each class.

Different features were found to specialize in different expressions; for example, the feature space RAW specializes in the neutral class, LPQ specializes in the angry and happy classes, LBP specializes in the disgust class, and HOG specializes in the surprise and sad classes, while the fear class did not have a predominant feature space.

We observed that the LPQ, HOG, and RAW methods were present in more than 80% of the selected combinations. This demonstrates that they play a significant role in the classification of some expressions.

We presented the most explanatory features for facial expression recognition. Feature relevance techniques appear to be some of the most-used schemes for post hoc explanations [4,5], as they are capable of providing an explicit description of the inner behavior of the model. There was no combination of features in a model that appeared to be clearly the best, leading to the conclusion that the recognition of a facial expression is the result of a combination of many features.

To the best of our knowledge, there has been no other study focused on the explainability of features. In this work, a system based on SRC was presented to provide an explanation, thus determining the most important features for each expression. This information can be useful in designing neural networks or performing fine-tuning in existing ones. CNN-based methods can be stripped from the top classification layer and that vector is, in fact, a feature representation, but we cannot have any knowledge of how these features are calculated and every time the CNN is trained, those features can change. Our method is based on well-known feature extraction methods; therefore, we can establish what features are more revealing. We can take advantage of this knowledge to improve the performance of all the models, including the CNN-based methods.

In our future work, we will explore the most focused-upon face regions for each feature vector, in order to determine the most interesting regions used for features and for each expression.

**Author Contributions:** Conceptualization, J.M.B.-R. and A.J.-i.-C.; methodology, P.D.M.-F., A.J.-i.-C. and T.I.R.; software, P.D.M.-F.; formal analysis, P.D.M.-F. and J.M.B.-R.; data curation, P.D.M.-F. and J.M.B.-R.; writing—original draft preparation, P.D.M.-F.; writing—review and editing, T.I.R. and A.J.-i.-C.; funding acquisition, A.J.-i.-C. All authors have read and agreed to the published version of the manuscript.

**Funding:** The research was funded by Project PID2019-104829RA-I00 "EXPLainable Artificial INtelligence systems for health and well-beING (EXPLAINING)" funded by MCIN/AEI/10.13039/501100011033.

**Institutional Review Board Statement:** Not applicable.

**Informed Consent Statement:** Not applicable.

**Data Availability Statement:** The data involved in the article are all shown in the figures and tables. However, they are also available from the first author on reasonable request.

**Conflicts of Interest:** The authors declare that they have no competing interests.

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
