# Peer review of "An Approach for Selecting the Most Explanatory Features for Facial Expression Recognition"

_applsci, doi:10.3390/app12115637_

Round 1

Reviewer 1 Report

Notes:

  • I like to see works exploring features and not the usual neural network approach;
  • It was missing a solid "Related Work" section with the needed subsections. With this section, probably the Introduction became better (less confusing) and more focused on introducing the paper;
  • Improve the article writing. Sometimes it gets confusing, namely in the results section;
  • Please illustrate better the methods and some results with the obtained images. A computer vision paper benefits from this possibility.

Abstract:

  • Please provide a measure of accuracy for the best features (example in percentage).

Introduction:

  • "...features extraction in FER.." - Explain what is FER;
  • "..SRC (Sparse Representation based Classifier)" - The correct way is first the Sparse Representation based Classifier and then "(SRC)";
  • It is a bit confusing since it is missing a proper "related work" section. This section could divide the applications into subsections improving the clarity of the paper;
  • Missing a description of the innovation. The innovation should be clear to the reader. You described the objective but not the innovation;
  • Missing a sentence describing the organization of the paper. "In Section XXX, it will be described .....".

Methodology:

  • Can you represent the methodology using a figure or a schematic? It will be better to understand it;
  • Some details of the well-known techniques like LBP are not necessary. Just cite the methods;
  • Describe the implementation details (e.g. used path size) in the Results section or Experiments. It will become more clear;
  • Since you did not introduce the innovation in the Introduction, it is not clear in the article what is "new" and what is "old";
  • Do not use acronyms without defining them e.g. Relu;
  • In Table 1 you define the acronyms a second time;
  • "..class $w_j$." ?.

Results:

  • Improve the Figures quality;
  • Described the used performance metrics in a proper subsection;
  • Please include "conclusions" in each subsection. It will resume the most important results;
  • Given "infinite time" I can combine infinite features to give me the best results I want. Processing time is critical. Please describe it;
  • It is missing some images or situations (the most important) where the features are very good and where they fail. A computer vision paper will benefit from this.

Conclusions:

  • You are missing an indication of the most important results, e.g. percentage or even accuracy.

Author Response

Reviewer #1:

Coauthors and I very much appreciated the encouraging, critical and constructive comments on this manuscript by the reviewer. The comments have been very thorough and useful in improving the manuscript. We strongly believe that the comments and suggestions have increased the scientific value of revised manuscript by many folds. We have taken them fully into account in revision. We are submitting the corrected manuscript with the suggestion incorporated the manuscript (you can see corrections with word change control). The manuscript has been revised as per the comments given by the reviewer, and our responses to all the comments are as follows:

Notes:

  • I like to see works exploring features and not the usual neural network approach;

Response- Thank you so much.

  • It was missing a solid "Related Work" section with the needed subsections. With this section, probably the Introduction became better (less confusing) and more focused on introducing the paper;

Response- Related Work is a new section.

  • Improve the article writing. Sometimes it gets confusing, namely in the results section;

Response- The paper has been edited (or will be edited) by MDPI Language Editing Services.

  • Please illustrate better the methods and some results with the obtained images. A computer vision paper benefits from this possibility.

Response- We added figure 3 with the result of different images.

Abstract:

  • Please provide a measure of accuracy for the best features (example in percentage).

Response- We did it.

We obtained the most explanatory features for each facial expression. We observed that LPQ (83%), HOG (82%) and RAW (82%) features are the most frequent to improve results, and some feature specializes in one expression (RAW for Neutral,  LPQ for Angry and Happy, LBP for Disgust, and HOG for Surprise).

Introduction:

  • "...features extraction in FER.." - Explain what is FER;

Response- We fixed it.

  • "..SRC (Sparse Representation based Classifier)" - The correct way is first the Sparse Representation based Classifier and then "(SRC)";

Response- We fixed it.

  • It is a bit confusing since it is missing a proper "related work" section. This section could divide the applications into subsections improving the clarity of the paper;

Response- Related Work is now a new section.

  • Missing a description of the innovation. The innovation should be clear to the reader. You described the objective but not the innovation;

Response- We included next text in line 132:

The innovation of this work is the exploration of the features extractors to select the most explanatory for facial expression recognition using sparse representation based classifier.

  • Missing a sentence describing the organization of the paper. "In Section XXX, it will be described .....".

Response- We fixed it. Line 52.

Methodology:

  • Can you represent the methodology using a figure or a schematic? It will be better to understand it;

Response-  We added figure 1 that represents the classification schema

  • Some details of the well-known techniques like LBP are not necessary. Just cite the methods;

Response- At this review round we will maintain the technics details, because the others reviewers did not make any comment about this. In case you maintain the recommendation, we will only cite and remove the details.

  • Describe the implementation details (e.g. used path size) in the Results section or Experiments. It will become more clear;

Response- From our point of view, we already included in line 338.

  • Since you did not introduce the innovation in the Introduction, it is not clear in the article what is "new" and what is "old";

Response- We included next text in line 132:

The innovation of this work is the exploration of the features extractors to select the most explanatory for facial expression recognition using sparse representation based classifier.

  • Do not use acronyms without defining them e.g. Relu;

Response- We fixed it and we reviewed all the acronyms.

  • In Table 1 you define the acronyms a second time;

Response- We fixed it and we only included the acronyms.

  • "..class $w_j$." ?.

Response- We fixed it

Results:

  • Improve the Figures quality;

Response- We improved Figure 2, Figure 3 and Figure 4 quality.

  • Described the used performance metrics in a proper subsection;

Response- We included subsection 4.3 Metrics in line 349. We consider that it is more suitable to include this subsection in Experiment section, because the metrics define the experiment.

  • Please include "conclusions" in each subsection. It will resume the most important results;

Response- For subsections 5.1 and 5.2 there were a summarize in the last paragraph of each subsection. Now, we remark that it is a summarize (line 443 and line 468). Regarding subsection 5.3, we included a summarize paragraph (line 507) as follows:

To sum up this subsection, LPQ, HOG and RAW methods are present in most of the selected combinations and some feature specializes in one expression (RAW for Neutral, LPQ for Angry and Happy, LBP for Disgust, and HOG for Surprise).

  • Given "infinite time" I can combine infinite features to give me the best results I want. Processing time is critical. Please describe it;

Response- We described it in new section 4.4 Experimental environment. We added follow text:

The hardware used to carry out computation is a desktop computer: CPU Intel i9-9900KF (16) @ 5.000GHz GPU NVIDIA GeForce RTX 2060 Memory: 7003MiB / 32035MiB OS: Ubuntu 20.04.4 LTS x86_64. With this configuration the total computing time was 12.3 hours (features extraction, sparse representation, training and metrics).

  • It is missing some images or situations (the most important) where the features are very good and where they fail. A computer vision paper will benefit from this.

Response- We added figure 3 with the result of different images.

Conclusions:

  • You are missing an indication of the most important results, e.g. percentage or even accuracy.

Response- We included as follows (line 516):

In all the cases, the combination of different features improves the results that using only one feature. Individually, the best feature is HoG (in 82% of combinations) and LPQ (in 83% of combinations). But, combining different features the best selection was RAW/GW/LBP/LPQ/GEO. Curiously, the best schemes (R/W/H/P/G/F+MP and R/W/B/P/G+MP accuracy 0.992, 0.828 and 0.776 respectively) combine the best and the worst individual schemes.

Reviewer 2 Report

This paper extracts local (GW, LBP, HOG, LPQ, RAW), geometric (GEO) and global (VGG, VGGFace) features for facial expression recognition. The objective is to see which of them give the best results when fed to a Sparse Representation based classifier. Authors create quite a large number of scenarios: 14 classifier rules for all feature type subsets. Evaluations are provided on datasets CK+, JAFFE and BU-3DFE.

There is very little variability in the results within one dataset. This means that neither the features nor the classifier rules have a significant impast on the result. Different aspects of the feature extraction have to be examined, for example the sparse representation or the whole range of new trainable feature extractors and classifiers.

There are also many formatting errors. Paper seems unfinished (line 502), Tables 2,3,5,6,9 are so large that they overflow to next pages, Figure 2 has separated caption from image, related work should include the respective author's names.

Algorithm 2:
Equation (5): what is "a" at the end?
Equations (5) and (9): replace "\in" with "=".
Equation (6): can you give a formula for "\delta"?

The work provides a large amount of experiments but I do not see a sufficient research contribution for this journal. If the authors select the most interesting experiments and evaluate more recent image classifiers, it would be a good paper for a conference.

Author Response

Coauthors and I very much appreciated the encouraging, critical and constructive comments on this manuscript by the reviewer. The comments have been very thorough and useful in improving the manuscript. We strongly believe that the comments and suggestions have increased the scientific value of revised manuscript by many folds. We have taken them fully into account in revision. We are submitting the corrected manuscript with the suggestion incorporated the manuscript (you can see corrections with word change control). The manuscript has been revised as per the comments given by the reviewer, and our responses to all the comments are as follows:

This paper extracts local (GW, LBP, HOG, LPQ, RAW), geometric (GEO) and global (VGG, VGGFace) features for facial expression recognition. The objective is to see which of them give the best results when fed to a Sparse Representation based classifier. Authors create quite a large number of scenarios: 14 classifier rules for all feature type subsets. Evaluations are provided on datasets CK+, JAFFE and BU-3DFE.

There is very little variability in the results within one dataset. This means that neither the features nor the classifier rules have a significant impast on the result. Different aspects of the feature extraction have to be examined, for example the sparse representation or the whole range of new trainable feature extractors and classifiers.

There are also many formatting errors. Paper seems unfinished (line 502), Tables 2,3,5,6,9 are so large that they overflow to next pages, Figure 2 has separated caption from image, related work should include the respective author's names.

Algorithm 2:
Equation (5): what is "a" at the end?

Response- It was a typo, we fixed it.

Equations (5) and (9): replace "\in" with "=".

Response- We fixed it. Regardless, although a lot of literature applies the “=” operator, mathematically the correct operator is "\in”, since the argmax operator returns a set, which can be the empty set, or even the entire evaluation set.

Equation (6): can you give a formula for "\delta"?

Response- Delta function is defined previously, in line 236

The work provides a large amount of experiments but I do not see a sufficient research contribution for this journal. If the authors select the most interesting experiments and evaluate more recent image classifiers, it would be a good paper for a conference.

Response- We consider that the work advances the state of the art in different aspects and is more than adequate to be published in this journal:

  1. As far as we know, there is no work where a similar study is carried out. All explainability studies focus on neural networks, on people, or on the data set, but not on features.
  2. From this study it is concluded that the SRC method is valid for selecting the feature extractor that most influences decision making. It is not only applicable to the FER, it can be applied to other decision-making systems.
  3. It is determined which characteristics are more important for the different expressions.
  4. This information is useful to use in the design and fine-tuning of neural networks for FER classification.
  5. The work explores features and not the usual neural network approach
  6. This research focus on the critical aspect of feature extraction, because extract the features are the most crucial step to have a good accuracy.

The other reviewers agreed on these advances.

Reviewer 3 Report

This research focus on all critical aspect of research, i.e., features; features are the most crucial step of any problem, if the practical set of features are extracted, the outcome accuracy will be on the higher side and vice versa. The research is very detailed and produced valuable insights about features extraction. Only a few minor modifications are required; it's good to be published afterward.

  1. The abstract is written ineffectively. Reading it gives a feeling of bits and pieces combined to make it. This needs to be rewritten.
  2. Acronyms must be written in full for the first time used, and later on, just abbreviations may be used. Correct this throughout the text.

Author Response

Coauthors and I very much appreciated the encouraging, critical and constructive comments on this manuscript by the reviewer. The comments have been very thorough and useful in improving the manuscript. We strongly believe that the comments and suggestions have increased the scientific value of revised manuscript by many folds. We have taken them fully into account in revision. We are submitting the corrected manuscript with the suggestion incorporated the manuscript (you can see corrections with word change control). The manuscript has been revised as per the comments given by the reviewer, and our responses to all the comments are as follows:

This research focus on all critical aspect of research, i.e., features; features are the most crucial step of any problem, if the practical set of features are extracted, the outcome accuracy will be on the higher side and vice versa. The research is very detailed and produced valuable insights about features extraction. Only a few minor modifications are required; it's good to be published afterward.

  1. The abstract is written ineffectively. Reading it gives a feeling of bits and pieces combined to make it. This needs to be rewritten.

Response- The abstract has been rewritten, also the paper has been edited (or will be edited) by MDPI Language Editing Services.

  1. Acronyms must be written in full for the first time used, and later on, just abbreviations may be used. Correct this throughout the text.

Response- We reviewed all the acronyms and we fixed.

Round 2

Reviewer 1 Report

The authors have addressed all my comments, and the manuscript is better.

It is a minimal contribution to the field and probably without the innovation needed to be considered for publication, but it can be regarded as one.

Still missing a comparison with other state-of-the-art methods.

Author Response

Coauthors and I very much appreciated the encouraging, critical and constructive comments on this manuscript by the reviewer. The comments have been very thorough and useful in improving the manuscript. We strongly believe that the comments and suggestions have increased the scientific value of revised manuscript by many folds. We have taken them fully into account in revision. We are submitting the corrected manuscript with the suggestion incorporated the manuscript (you can see corrections with MS WORD change control, please we want to remind that in PDF article filetype change control are not visible). The manuscript has been revised as per the comments given by the reviewer, and our responses to all the comments are as follows:

Comments and Suggestions for Authors

The authors have addressed all my comments, and the manuscript is better.

It is a minimal contribution to the field and probably without the innovation needed to be considered for publication, but it can be regarded as one.

Response: Thank you very much for your comments and suggestions, that help us to improve our work.

Still missing a comparison with other state-of-the-art methods.

Response: As far as we know, there are not works related to explicability for facial expression recognition. In fact, for comparison purpose and from 2 years ago, we are working on XAI for FER and we did not find any related work in this topic

Reviewer 2 Report

I see that Authors made some changes but the reviewing the updated version is difficult because I do not see the changes highlighted.

There are so many typesetting errors that Authors should consider some LaTeX assistance.
Fix the page-separated Table 4 and Algorithm 2.
Don't use "*" as multiplication sign.
Use horizontal spacing between "s.t."
Lines 12-13 is the default style text?
Remove citations from abstract.
Add author's names to citations in related work.

Equation (7): ?.,? refers to the values of all columns  -- do you mean that it refers to the column j?

Equations (5) and (9): replace "\in" with "=".
Response- We fixed it. Regardless, although a lot of literature applies the “=” operator, mathematically the correct operator is "\in”, since the argmax operator returns a set, which can be the empty set, or even the entire evaluation set.
Reviewer: Equations (5) and (9) are definitions, correct? Normally a definition sign can be "=" "\leftarrow" or ":=", but I have never seen "\in".

Equation (6): can you give a formula for "\delta"?
Response- Delta function is defined previously, in line 236
Reviewer: No, line 236 defines a set of training signals D.

The work provides a large amount of experiments but I do not see a sufficient research contribution for this journal. If the authors select the most interesting experiments and evaluate more recent image classifiers, it would be a good paper for a conference.
Response- We consider that the work advances the state of the art in different aspects and is more than adequate to be published in this journal: [6 points]. The other reviewers agreed on these advances.
Reviewer: Add a random classifier for comparison.
Also, if for one given classifier a set of features gives better results than other set of features, it doesn't necessarily mean that they are always better. They might be worse with another given classifier. That is why I suggested experiments on various types of classifiers.

No other relevant method is being compared against.
Opinion of other reviewers is
 to me: [1] unknown and [2] irrelevant.

Current version is still a reject. Please make the changes and I will have a look at the revised version.

Author Response

Coauthors and I very much appreciated the encouraging, critical and constructive comments on this manuscript by the reviewer. The comments have been very thorough and useful in improving the manuscript. We strongly believe that the comments and suggestions have increased the scientific value of revised manuscript by many folds. We have taken them fully into account in revision. We are submitting the corrected manuscript with the suggestion incorporated the manuscript (you can see corrections with MS WORD change control, please we want to remind that in PDF article filetype change control are not visible). The manuscript has been revised as per the comments given by the reviewer, and our responses to all the comments are as follows:

Comments and Suggestions for Authors

I see that Authors made some changes but the reviewing the updated version is difficult because I do not see the changes highlighted.

Response: We used MS Word as document editor. For submitting the revision, we followed the instructions of the editorial, and we enabled the track changes. They indicated:

(II) Any revisions to the manuscript should be marked up using the “Track

Changes” function if you are using MS Word/LaTeX, such that any changes can

be easily viewed by the editors and reviewers.”

We downloaded the document Manuscript File and we checked that changes are tracked.

There are so many typesetting errors that Authors should consider some LaTeX assistance.

Response: We used MS Word.

Fix the page-separated Table 4 and Algorithm 2.

Response: From our experience, if the paper is accepted MDPI will format the document in a proper way.

Don't use "*" as multiplication sign.

Response: We downloaded the Manuscript File, and with our version of MS Word we do not find any * as a multiplication sign. If you can indicate us the line, table, figure or equation we will fix it.

Use horizontal spacing between "s.t."

Response: We fixed.

Lines 12-13 is the default style text?

Response: It was a typo. We deleted it.

Remove citations from abstract.

Response: We did it.

Add author's names to citations in related work.

Response: We did it.

Equation (7): ?.,? refers to the values of all columns  -- do you mean that it refers to the column j?

Response: We changed it:
“where  refers to the column  as a vector and  represents the decision profile”

Equations (5) and (9): replace "\in" with "=".
Response- We fixed it. Regardless, although a lot of literature applies the “=” operator, mathematically the correct operator is "\in”, since the argmax operator returns a set, which can be the empty set, or even the entire evaluation set.
Reviewer: Equations (5) and (9) are definitions, correct? Normally a definition sign can be "=" "\leftarrow" or ":=", but I have never seen "\in".

Response: We already replaced "\in" with "=" in round 1.

Equation (6): can you give a formula for "\delta"?
Response- Delta function is defined previously, in line 236
Reviewer: No, line 236 defines a set of training signals D.

Response: It was a typo in the line number. Line 243. We defined as follow:

“where  is a vector of coefficients having many values equal to zero, except for those associated with the class .”

The work provides a large amount of experiments but I do not see a sufficient research contribution for this journal. If the authors select the most interesting experiments and evaluate more recent image classifiers, it would be a good paper for a conference.
Response- We consider that the work advances the state of the art in different aspects and is more than adequate to be published in this journal: [6 points]. The other reviewers agreed on these advances.
Reviewer: Add a random classifier for comparison.
Also, if for one given classifier a set of features gives better results than other set of features, it doesn't necessarily mean that they are always better. They might be worse with another given classifier. That is why I suggested experiments on various types of classifiers.
No other relevant method is being compared against.
Opinion of other reviewers is to me: [1] unknown and [2] irrelevant.

Current version is still a reject. Please make the changes and I will have a look at the revised version.

Response: As we indicated in the paper, previous work shows that SR have good results for FER (line 102):

…The output of each classifier was used to evaluate a decision rule, and they applied combination rules [23]. Of these, Product Rule (PR) and the SR provided the best results. Several studies have also employed dynamic features [24]–[26]. In these works, the variability of facial changes was studied using regions or points of interest in the face image…

Moreover, in this second round we included in our paper (line 467) the reference Ramis et al. [65], where the authors analyze the results of classification using different datasets. They trained a neural network with BU-4FDE and test with JAFFE obtaining an accuracy of 0.4317. That is the same experiment with our classifiers, and we obtained and accuracy of 0.547 (table 9 of our paper). Please, notice that our classification results are similar (better with this network of paper [65]). Remark, that neural networks are a black box architecture and cannot be (or are very hard) to explain. Then make sense to study the most explanatory features with the classifier that according to state of the art provides good results for FER. It will not have sense to not to analyze the most explanatory features with classifiers with worse accuracy, because that will mean that features would be worse for FER.

We included next text in line 467:

Remark that Ramis et al. [65] also trained with BU-4FDE and test with JAFFE obtaining an accuracy of 0.4317.

[65] Ramis, S. et al. “A Novel Approach to Cross dataset studies in Facial Expression Recognition”. Multimedia Tools and Applications (2022). https://doi.org/10.1007/s11042-022-13117-2

Round 3

Reviewer 2 Report

Authors made a noticeable improvement in typesetting and reformulated the critical parts of the paper for a better understanding.

They also added a related method [65] for comparison, but it is a bit buried in the middle of a paragraph. It is indicated that their method is rather similar. Please explain how it differs from your method.

Reviewer: Don't use "*" as multiplication sign.
Authors: We downloaded the Manuscript File, and with our version of MS Word we do not find any * as a multiplication sign. If you can indicate us the line, table, figure or equation we will fix it.
Reviewer: Captions of Tables 6,7,8 have the formula (?∗1000^{−1}).

To my understanding, the main objective of the paper is to evaluate which features (or their combinations) are the most appropriate for face expression recognition. There are numerous methods published on face expression recognition and each of them use features in some form. For example, any CNN-based method can be stripped from the top classification layer and that vector is in fact a feature representation. Are such features worth evaluating? Please explain that somewhere in the text.

Weak accept after the last changes are made.

Author Response

Coauthors and I very much appreciated the encouraging, critical and constructive comments on this manuscript by the reviewer. The comments have been very thorough and useful in improving the manuscript. We strongly believe that the comments and suggestions have increased the scientific value of revised manuscript by many folds. We have taken them fully into account in revision. We are submitting the corrected manuscript with the suggestion incorporated the manuscript (you can see corrections with MS WORD change control, please we want to remind that in PDF article filetype change control are not visible). The manuscript has been revised as per the comments given by the reviewer, and our responses to all the comments are as follows:

Comments and Suggestions for Authors

Authors made a noticeable improvement in typesetting and reformulated the critical parts of the paper for a better understanding.

They also added a related method [65] for comparison, but it is a bit buried in the middle of a paragraph. It is indicated that their method is rather similar. Please explain how it differs from your method.

Response: We extended the explanation in section 5.1 with the next paragraph:

In Ramis et al. [65] present a method to combine multiple datasets and conduct an exhaustive evaluation of a proposed system based on a CNN analyzing and comparing performance using single and cross-dataset approaches with other architectures. Remark that they trained with BU-4FDE and test with JAFFE obtaining an accuracy of 0.4317. In our proposal, we study the most important features by applying different feature extraction methods for facial expression representation, transforming each obtained representation into a Sparse Representation (SR) domain, and train combination models to classify signals, using the Extended Cohn–Kanade (CK+), BU-3DFE, and JAFFE data sets for validation.

Reviewer: Don't use "*" as multiplication sign.
Authors: We downloaded the Manuscript File, and with our version of MS Word we do not find any * as a multiplication sign. If you can indicate us the line, table, figure or equation we will fix it.
Reviewer: Captions of Tables 6,7,8 have the formula (?∗1000^{−1}).

Response: We fixed it.

To my understanding, the main objective of the paper is to evaluate which features (or their combinations) are the most appropriate for face expression recognition. There are numerous methods published on face expression recognition and each of them use features in some form. For example, any CNN-based method can be stripped from the top classification layer and that vector is in fact a feature representation. Are such features worth evaluating? Please explain that somewhere in the text.

Response:  We added the following paragraph in conclusion section.

CNN-based methods can be stripped from the top classification layer and that vector is in fact a feature representation, but we cannot have any knowledge on how these features are calculated and every time the CNN is trained features can change. Our method is based in well-known feature extraction methods, therefore we can know what features are more explanatories. We can take the advantage of this knowledge to improve the performance of the models, including the CNN-based methods.